# Dissociable memory modulation mechanisms facilitate fear amnesia at different timescales

Yinmei Ni[1†], Ye Wang[1†], Zijian Zhu[2], Jingchu Hu[3], Daniela Schiller[4], Jian Li[1,5*]

[1]School of Psychological and Cognitive Sciences and Beijing Key Laboratory of Behavior and Mental Health, Peking University, Beijing, China; [2]School of Psychology, Shaanxi Normal University, Xi'an, China; [3]Department of Anxiety Disorders, Shenzhen Kangning Hospital, Shenzhen Mental Health Institute, Shenzhen, China; [4]Departments of Psychiatry and Neuroscience, and Friedman Brain Institute, Icahn School of Medicine at Mount Sinai, New York, United States; [5]IDG/McGovern Institute for Brain Research, Peking University, Beijing, China

*For correspondence:
leekin@gmail.com

[†]These authors contributed equally to this work

Competing interest: The authors declare that no competing interests exist.

## eLife Assessment

This manuscript presents **valuable** findings which reveal an intricate pattern of memory expression following retrieval extinction at different intervals from retrieval-extinction to test. The novel advance is in the demonstration that, relative to a standard extinction procedure, the retrieval-extinction procedure more effectively suppresses responses to a conditioned threat stimulus when testing occurs just minutes after extinction. While the data provide **solid** evidence that the "short-term" suppression of responding involves engagement of the dorsolateral prefrontal cortex, there are inconsistencies in the analyses reported which obscure the interpretation and leave some of the claims with limited evidence.

**Abstract** Memory reactivation renders consolidated memory fragile and thereby opens the window for memory updates, such as memory reconsolidation. However, whether memory retrieval facilitates update mechanisms other than memory reconsolidation remains unclear. We tested this hypothesis in three experiments with healthy human participants. First, we demonstrate that memory retrieval-extinction protocol prevents the return of fear expression shortly after extinction training, and this short-term effect is memory reactivation dependent (Study 1, N=57 adults). Furthermore, across different timescales, the memory retrieval-extinction paradigm triggers distinct types of fear amnesia in terms of cue specificity and cognitive control ability dependence, suggesting that the short-term fear amnesia might be caused by different mechanisms from the cue-specific amnesia at a longer and separable timescale (Study 2, N=79 adults). Finally, using continuous theta-burst stimulation (Study 3, N=75 adults), we directly manipulated brain activity in the dorsolateral prefrontal cortex and found that both memory reactivation and intact prefrontal cortex function were necessary for the short-term fear amnesia after the retrieval-extinction protocol. The differences in temporal scale, cue specificity, and cognitive control ability dependence between the short- and long-term amnesia suggest that memory retrieval and extinction training trigger distinct underlying memory update mechanisms. These findings suggest the potential involvement of coordinated memory modulation processes upon memory retrieval and may inform clinical approaches for treating persistent maladaptive memories.

## Introduction

Memory retrieval provides a unique window through which consolidated memories can be modified or even neutralized. Studies of memory reconsolidation suggest that memories are rendered fragile each time they are retrieved (*Alberini, 2005*; *Dudai, 2006*; *Nader et al., 2000*), possibly due to the disruption of the old memory trace. The reconsolidation of the new memory trace also requires de novo protein synthesis, which usually takes hours to complete. Pharmacological blockade of protein synthesis and behavioral interventions diminish the original fear memory expression in the long-term (24 hr later) memory test (*Lee, 2008*; *Lee et al., 2017*; *Schiller et al., 2013*; *Schiller et al., 2010*), resulting in the cue-specific fear memory deficit (*Debiec et al., 2002*; *Lee, 2008*; *Nader et al., 2000*). For example, during the reconsolidation window, retrieving a fear memory allows it to be updated through extinction training (i.e. the retrieval-extinction paradigm; *Agren, 2014*; *Agren et al., 2012*; *Lee, 2008*; *Lee et al., 2017*; *Schiller et al., 2013*; *Schiller et al., 2010*, but also see *Chalkia et al., 2020a*; *Chalkia et al., 2020b*; *Schiller et al., 2020*).

However, the exact functional role of memory retrieval remains unclear. For example, while previous fear memory studies typically reported the reconsolidation effects that emerged several hours later (the long-term effect) (*Kindt and Soeter, 2018*; *Lee, 2008*; *Speer et al., 2021*), the temporal dynamics of the amnesia effect through behavioral manipulation such as retrieval-extinction was often overlooked in the fear memory literature (*Lee et al., 2017*; *Schiller et al., 2013*; *Schiller et al., 2010*). It is possible that memory retrieval might facilitate memory modification at different temporal scales. The long-term effect is in stark contrast to semantic and episodic memory studies where the behavioral manipulations such as the retrieval-induced forgetting (RIF) protocol or direct suppression result in short-term memory deficit within 30 min (*Anderson et al., 2000*; *Anderson et al., 1994*). More recently, behavioral strategies originally developed in the declarative memory studies such as direct suppression had also been successfully applied to the fear memory paradigms in humans and prevented the return of fear shortly afterward (30 min), suggesting that the fear memory might be amenable to modulation at a more immediate timescale, to which the memory reconsolidation theory remains agnostic (*Wang et al., 2021*).

Corresponding to the long-term behavioral manifestation, concurrent neural evidence supporting memory reconsolidation hypothesis emphasizes that synapse degradation and de novo protein synthesis are required for fear memory reconsolidation. These processes typically take several hours, if not longer, and are often aided by overnight sleep (*Kindt and Soeter, 2018*). On the contrary, previous behavioral manipulations engendering the short-term amnesia on declarative memory, such as the think/no-think (TNT) paradigm, hinge on the intact activities in brain areas such as dorsolateral prefrontal cortex (cognitive control) and its functional coupling with specific brain regions such as hippocampus (memory retrieval) (*Anderson and Green, 2001*; *Wimber et al., 2015*). The declarative amnesia effect arises much earlier due to the more instant modulation of functional connectivity, rather than the slower processes of new protein synthesis in these brain regions (*Wang et al., 2021*). However, it remains unclear whether experimental paradigms such as memory retrieval-extinction, a method developed to induce long-term prevention of fear expression based on the memory reconsolidation hypothesis, might also trigger short-term amnesia for the fear memory.

To fully comprehend the temporal dynamics of the memory retrieval-extinction training effect, we first carried out a simple fear memory study by pairing a neutral cue (conditioned stimulus [CS]) with the electric shock (unconditioned stimulus [US]) to directly test whether the fear retrieval-extinction paradigm would yield a short-term effect on fear expression. We then performed another double cue CS-US fear memory experiment to study the temporal characteristics of fear memory malleability to behavioral extinction at the short- (30 min), medium- (6 hr), and long-term (24 hr) timescales.

We hypothesize that the labile state triggered by the memory retrieval may facilitate different memory deficits following extinction training, and these deficits can be further disentangled through the lens of temporal dynamics and cue specificities. In theory, different cognitive mechanisms underlying specific fear memory deficits, therefore, can be inferred based on the difference between memory deficits. Specifically, memory reconsolidation effect, if it exists, should only be evident in the long-term (24 hr) memory test due to the requirement of new protein synthesis and should be cue-dependent (*Monfils et al., 2009*). In contrast, if more immediate memory update mechanisms are involved, then the fear memory deficit should be observed relatively early (30 min) after extinction training (*Anderson and Floresco, 2022*; *Anderson and Green, 2001*). There may even exist a

'limbo' state: when the immediate memory update effects dissipate and the reconsolidation updating is yet to take effect (6 hr), the CSs should still elicit fear responses, regardless of the reminder (cue). Previous research in the short-term declarative and fear memory deficits highlighted the role of cognitive control. Subjects with stronger control over intrusive thoughts or stronger functional connectivity between brain regions such as the dorsolateral prefrontal cortex (dlPFC) and hippocampus exhibited more pronounced amnesia in the memory suppression tasks (*Anderson et al., 1994*; *Wells and Davies, 1994*; *Wimber et al., 2015*). We reason that if the retrieval-related short-term fear amnesia is also related to cognitive control and the dlPFC activities, then the individual difference of such immediate fear memory deficits should be associated with individual subject's control abilities over intrusive thoughts (*Küpper et al., 2014*), which can be measured via the Thought-Control Ability Questionnaire (TCAQ) scale.

Indeed, through a series of experiments, we identified a short-term fear amnesia effect following memory retrieval-extinction training, in addition to the long-term amnesia effect that appeared much later. Finally, by applying the continuous theta-burst stimulation (cTBS) protocol over the dlPFC, the brain region critical for cognitive control and goal maintenance, we showed that both memory reactivation and intact dlPFC activity were necessary for the more immediate fear amnesia effect.

## Results
### The short-term effect of the fear memory retrieval-extinction procedure

To test the memory-retrieval-induced fear memory deficits at different timescales in humans, we designed two experiments (Studies 1 and 2) to examine whether the fear memory retrieval-extinction procedure would block the return of fear memory.

In the first study, we aimed to test whether there is a short-term amnesia effect following the fear retrieval-extinction paradigm. We measured subjects' skin conductance responses (SCRs) for the fear acquisition, extinction, and test phases to assess the associative fear memory and the recovery of fear memory. At each phase, the differential fear response was calculated as the difference between SCRs to the CS+ and CS−.

To assess fear acquisition across groups (*Figure 1b and c*), we conducted a mixed two-way ANOVA of group (reminder vs. no-reminder) × time (early vs. late part of the acquisition; first 5 and last 5 trials, correspondingly) on the differential fear SCR. Our results showed a significant main effect of time (early vs. late; $F_{1,55} = 6.545$, p=0.013, $\eta^2$=0.106), suggesting successful fear acquisition in both groups. There was no main effect of group (reminder vs. no-reminder) or the group × time interaction (group: $F_{1,55} = 0.057$, p=0.813, $\eta^2$=0.001; interaction: $F_{1,55} = 0.066$, p=0.798, $\eta^2$=0.001), indicating similar levels of fear acquisition between two groups. Post hoc *t*-tests confirmed that the fear responses to the CS+ were significantly higher than that of CS− during the late part of the acquisition phase in both groups (reminder group: $t_{29}$=6.642, p<0.001; no-reminder group: $t_{26}$=8.522, p<0.001; *Figure 1c*). Importantly, the levels of acquisition were equivalent in both groups (early acquisition: $t_{55}$=−0.063, p=0.950; late acquisition: $t_{55} = -0.318$, p=0.751; *Figure 1c*).

To ensure equivalent and successful extinction across groups (*Figure 1b and d*), we performed similar mixed two-way ANOVA (group × trial) on the differential SCR in the extinction phase. Our results showed a significant main effect of trial ($F_{10,550} = 2.825$, p=0.010, $\eta^2$=0.049), but no effect of group ($F_{1,55} = 0.063$, p=0.802, $\eta^2$=0.001) or their interaction ($F_{10,550} = 1.77$, p=0.101, $\eta^2$=0.031). Follow-up *t*-tests further confirmed that in the last trial of extinction, there was no difference between fear responses to CS+ and CS− within group (reminder group: $t_{29}$=−1.147, p=0.261; no-reminder group: $t_{26}$=0.261, p=0.796; *Figure 1d*). Importantly, the levels of extinction did not show significant differences between the two groups ($t_{55} = -1.074$, p=0.288; *Figure 1d*).

Fear recovery was assessed using a two-way ANOVA with main effects of group (reminder vs. no-reminder) and time (last trial of extinction vs. first trial of test) and it showed a significant time × group interaction ($F_{1,55} = 4.087$, p=0.048, $\eta^2$=0.069). We defined the fear recovery index as the SCR difference between the first test trial and the last extinction trial for a specific CS. The interaction effect was confirmed by the significant difference between the differential fear recovery indexes between CS1+ and CS2+ in the reminder and no-reminder groups ($t_{55}$=−2.022, p=0.048; *Figure 1e*). Post hoc *t*-tests showed that fear memories were resilient after regular extinction training, as demonstrated by the

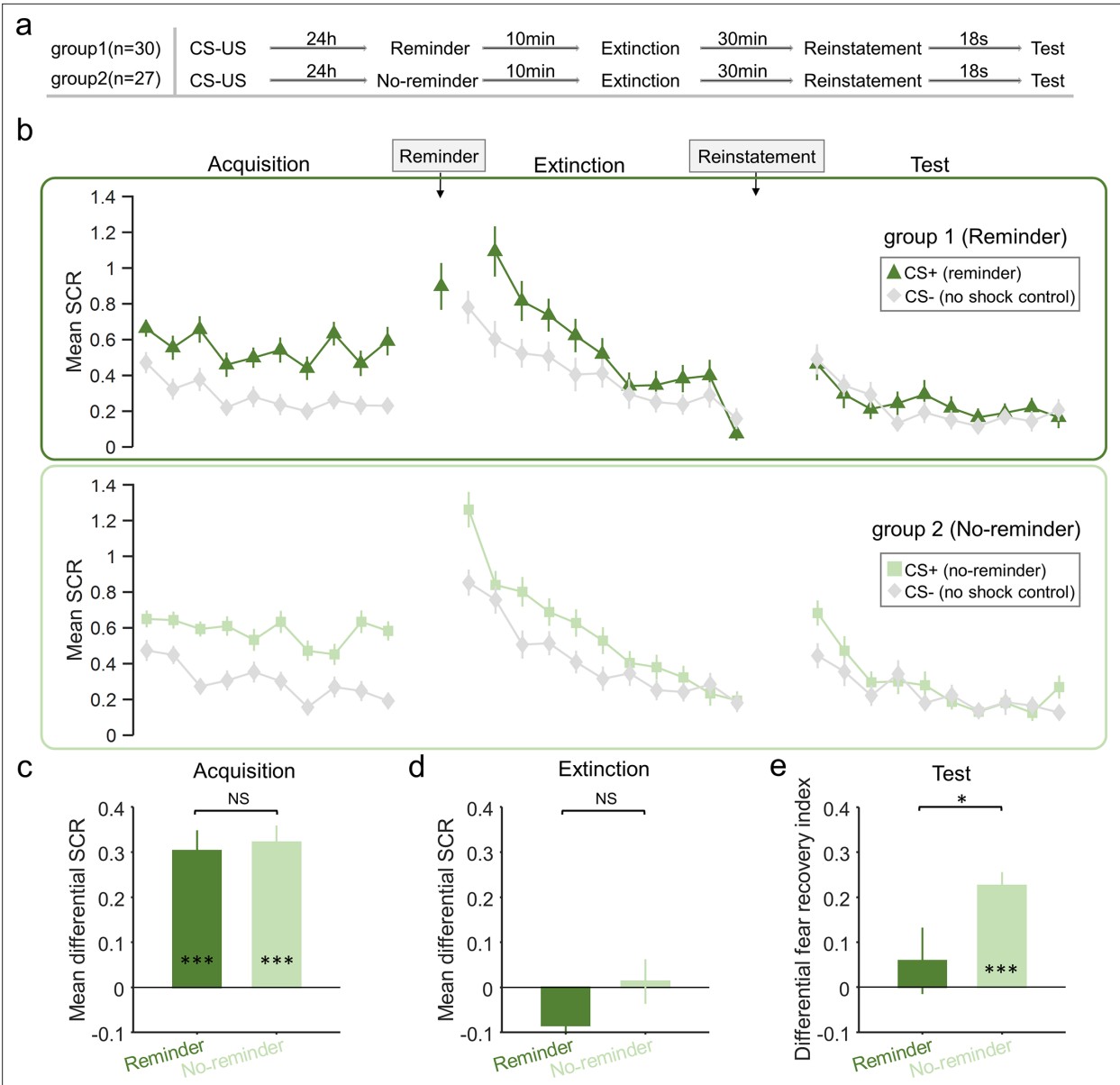

**Figure 1.** Skin conductance response (SCR) responses to conditioned stimuli in two groups of Study 1 indicating short-term memory amnesia. (**a**) Experimental design and timeline. (**b**) SCRs of fear-conditioned stimuli (CS+) and the control stimulus (CS–) across fear acquisition, extinction, and test phases for two groups (reminder and no-reminder). (**c**) Mean differential SCRs (CS+ minus CS–) in the acquisition phase (trials in the latter half). (**d**) Differential SCRs (CS+ minus CS–) in the extinction phase (last trial). (**e**) Differential fear recovery index between CS+ and CS– in the test phase. ***p<0.001, *p<0.05. NS: nonsignificant. Two sample t-test, error bars represent standard errors (n = 30 & 27 for the reminder and no-reminder groups, respectively).

The online version of this article includes the following figure supplement(s) for figure 1:

**Figure supplement 1.** Skin conductance response (SCR) responses to conditioned stimuli in two groups of Study 1 with all responders (learners+non-learners).

significant difference between fear recovery indexes of the CS+ and CS– for the no-reminder group ($t_{26}$=7.441, p<0.001; Bonferroni correction, *Figure 1e*), while subjects in the reminder group showed no difference of fear recovery between CS+ and CS– ($t_{29}$=0.797, p=0.432, *Figure 1e*). Also, our results remain robust even with the 'non-learners' included in the analysis (*Figure 1—figure supplement 1*).

Together, these results indicate that the retrieval-extinction procedure prevents the return of fear expression in the short-term test, a phenomenon distinct from the hypothesized fear reconsolidation effect at a much later time (*Schiller et al., 2013*). In the current study, fear recovery was tested 30 min

after extinction training, whereas the effect of memory reconsolidation was generally evident only several hours later and possibly with the help of sleep (*Kindt and Soeter, 2018*; *Speer et al., 2021*), leaving open the possibility of a different cognitive mechanism for the short-term fear amnesia related to the retrieval-extinction procedure. Importantly, such a short-term effect is also retrieval dependent, suggesting the labile state of memory may be necessary for the short-term memory update to take effect (*Figure 1e*).

## Temporal and generalizing characteristics of the retrieval-extinction effect

Given the findings of short-term fear amnesia following the retrieval-extinction paradigm, we set out to map the temporal dynamics of fear amnesia, as well as its cue specificities – whether the amnesia triggered by a specific CS+ reminder (e.g. CS1+) generalizes to the fear memory associated with the other CS+ (CS2+). We adopted a double-cue fear learning paradigm and examined fear recovery in the short- (30 min), medium- (6 hr), and long-term (24 hr) fear memory tests in three separate groups of participants after the extinction training (*Figure 2a*).

Fear acquisition was assessed using a mixed three-way ANOVA with group (30 min, 6 hr and 24 hr), CS+ (CS1+ vs. CS2+, defined as the mean SCR differences between each CS+ and CS–) and trial numbers (10 trials) as main factors. This analysis showed a significant main effect of trial ($F_{9,684}$ = 12.782, p<0.001, $\eta^2$=0.144), but no effect of group ($F_{2,76}$ = 2.121, p=0.127, $\eta^2$=0.053), CS+ ($F_{1,76}$ = 0.576, p=0.45, $\eta^2$=0.008) or their interactions (all Ps>0.05; *Figure 2b*). Furthermore, to ensure equivalent fear acquisition across three groups, a mixed two-way ANOVA showed no effect of group (30 min, 6 hr, and 24 hr; $F_{2,76}$ = 1.79, p=0.174, $\eta^2$=0.045), CS+ (CS1+ vs. CS2+; $F_{1,76}$ = 1.337, p=0.251, $\eta^2$=0.017), or their interaction ($F_{2,76}$ = 0.959, p=0.388, $\eta^2$=0.025) on the late phase of acquisition (last 5 trials). Post hoc *t*-tests confirmed that the SCRs elicited by CS1+ and CS2+ were significantly higher than that of CS– among three groups on the late phase of acquisition (CS1+: $t_{26}$=6.806, p<0.001; CS2+: $t_{26}$=6.854, p<0.001 for the 30 min group; CS1+: $t_{25}$=9.822, p<0.001; CS2+: $t_{25}$=9.841, p<0.001 for the 6 hr group and CS1+: $t_{25}$=6.909, p<0.001; CS2+: $t_{25}$=6.956, p<0.001 for the 24 hr group), indicating successful and similar fear acquisition across all groups (*Figure 2c*).

We then set out to examine whether participants underwent successful extinction training on day 2. The similar mixed three-way ANOVA showed a significant effect of trial ($F_{10,760}$ = 8.220, p<0.001, $\eta^2$=0.098), but no effect of group ($F_{2,76}$ = 1.822, p=0.169, $\eta^2$=0.046), CS+ ($F_{1,76}$ = 0.91, p=0.343, $\eta^2$=0.012) or their interactions (all Ps>0.1; *Figure 2b*). To test whether participants across all groups achieved similar levels of extinction, a mixed two-way ANOVA showed no effect of group (30 min, 6 hr, and 24 hr; $F_{2,76}$ = 0.059, p=0.943, $\eta^2$=0.002), CS+ (CS1+ vs. CS2+; $F_{1,76}$ = 0.258, p=0.613, $\eta^2$=0.003) or their interaction ($F_{2,76}$ = 0.504, p=0.606, $\eta^2$=0.013) on the last trial of extinction training. Post hoc *t*-tests showed that there was no difference between CS+ and CS– responses in all groups on the last trial of extinction (CS1+: $t_{26}$=–0.397, p=0.695; CS2+: $t_{26}$=–0.505, p=0.618 for the 30 min group; CS1+: $t_{25}$=–0.054, p=0.957; CS2+: $t_{25}$=0.140, p=0.890 for the 6 hr group and CS1+: $t_{25}$=0.435, p=0.668; CS2+: $t_{25}$=–0.317, p=0.754 for the 24 hr group; *Figure 2d*).

To assess the effects of the retrieval-extinction procedure as a function of the time delay between fear extinction and test phases, we conducted a three-way ANOVA with the within-subject factor CS+ (CS1+ vs. CS2+, defined by the mean SCR differences between each CS+ and CS–), time (last trial of extinction vs. first trial of test), and between-subject factor group (30 min, 6 hr, and 24 hr) and found a significant three-way interaction ($F_{2,76}$ = 6.376, p=0.003, $\eta^2$=0.144). To disentangle this interaction, we further examined the effects in each group.

In the 30 min group, there was no significant effect of time ($F_{1,26}$ = 0.208, p=0.652, $\eta^2$=0.008), CS+ ($F_{1,26}$ = 0.888, p=0.355, $\eta^2$=0.033), or their interaction ($F_{1,26}$ = 1.039, p=0.318, $\eta^2$=0.038). Post hoc *t*-tests showed that the retrieval-extinction training diminished fear responses to both the reminded CS+ (differential fear recovery index between CS1+ and CS–, $t_{26}$=–0.787, p=0.438; *Figure 2c*) and the non-reminded CS+ (differential fear recovery index between CS2+ and CS–, $t_{26}$=0.088, p=0.93; *Figure 2e*), suggesting a cue-independent short-term amnesia effect. This result indicates that the short-term amnesia effect observed in Study 2 is not reminder-cue specific and can generalize to the non-reminded cues. In addition, there was no significant difference between the fear recovery index associated with CS1+ and CS2+ ($t_{26}$=–1.019, p=0.318; *Figure 2e*).

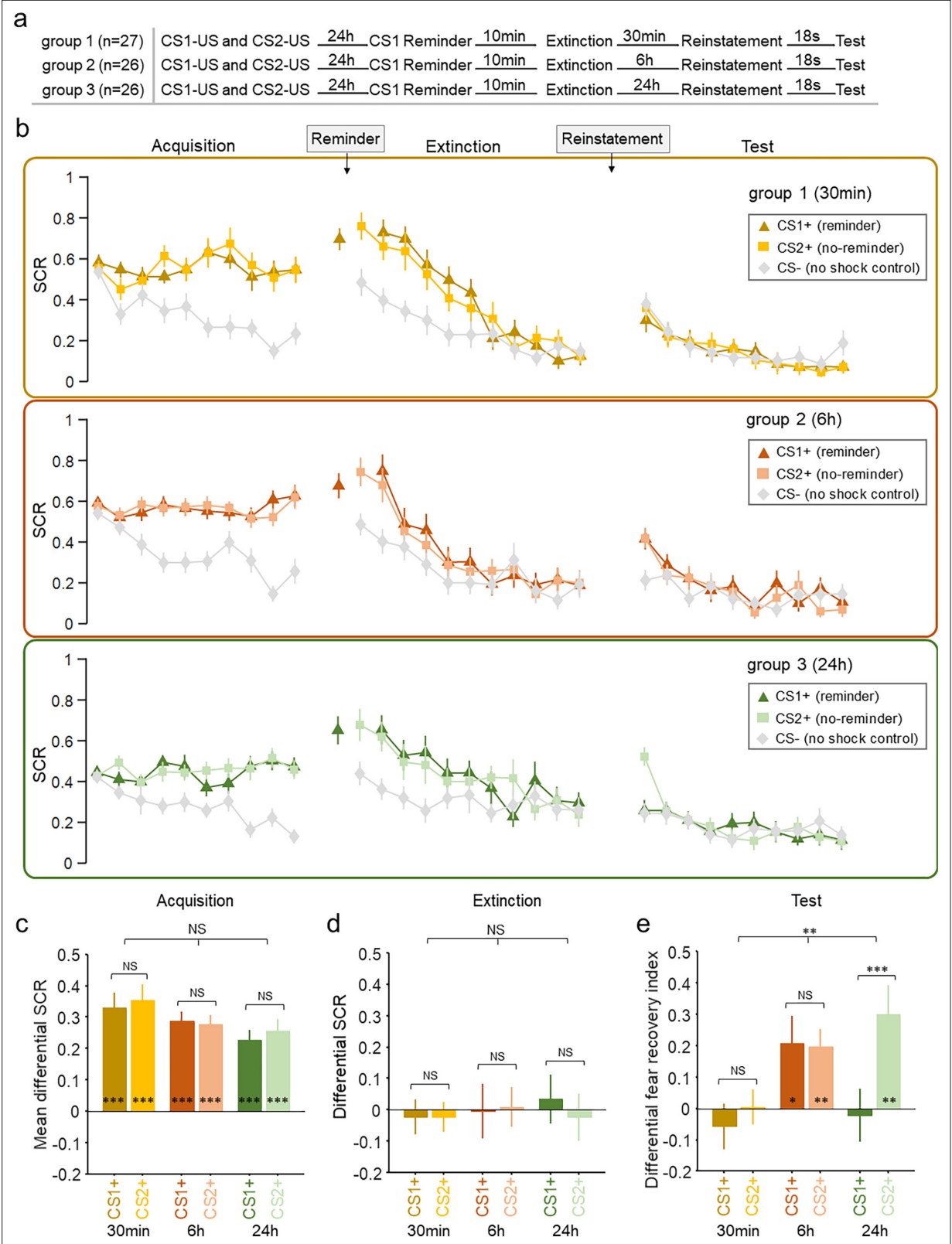

**Figure 2.** Skin conductance response (SCR) responses to conditioned stimuli in three groups of Study 2 indicating amnesia at different timescales. (a) Experimental design and timeline. (b) SCRs of fear-conditioned stimuli CS1+ (reminder) and CS2+ (no-reminder), and the control stimulus (CS–) across the fear acquisition, extinction, and test phases for each group (30 min, 6 hr and 24 hr groups). (c) Mean differential SCRs (CS1+ minus CS– and CS2+ minus CS–) in the acquisition phase (trials in the latter half). (d) Differential SCRs in the extinction phase (last trial). (e) Differential fear recovery

*Figure 2 continued on next page*

*Figure 2 continued*

index between CS+ and CS– in the test phase. \*\*\*p<0.001. \*\*p<0.01. \*p<0.05. NS: nonsignificant. ANOVA and post-hoc t-test, error bars represent standard errors (n = 27, 26 and 26 for the 30min, 6h and 24h groups, correspondingly).

The online version of this article includes the following figure supplement(s) for figure 2:

**Figure supplement 1.** Skin conductance response (SCR) responses to conditioned stimuli in three groups of Study 2 with all responders (learners+non-learners).

In contrast to the short-term effect, there was a significant effect of time ($F_{1,25}$ = 9.654, p=0.005, $\eta^2$=0.279), but no effect of CS+ ($F_{1,25}$ = 0.037, p=0.85, $\eta^2$=0.001) or their interaction ($F_{1,25}$ = 0.028, p=0.869, $\eta^2$=0.001) in the 6 hr (medium-term) group. Post hoc *t*-tests showed that fear memory recovered for the reminded CS+ (differential fear recovery index between CS1+ and CS–, $t_{25}$=2.382, p=0.025), as well as the non-reminded CS+ (differential fear recovery index between CS2+ and CS–, $t_{25}$=3.525, p=0.002) in the medium-term test of the retrieval-extinction procedure effect, suggesting the failure to suppress the return of extinguished fear memory at such a timescale (***Figure 2e***). There was also no significant difference measured by the fear recovery index between CS1+ and CS2+ ($t_{25}$=0.166, p=0.870; ***Figure 2e***).

Finally, in line with previous research on fear memory reconsolidation, significant SCR difference of the fear reinstatement effect between the reminded CS+ and non-reminded CS+ (time × CS+ interaction: $F_{1,25}$ = 16.924, p<0.001, $\eta^2$=0.404) was detected in the 24 hr group via two-way ANOVA. More specifically, the retrieval-extinction procedure only diminished fear responses to the reminded CS+ (differential fear recovery index between CS1+ and CS–, $t_{25}$=–0.25, p=0.805), whereas the fear response to the non-reminded CS+ remained significant (differential fear recovery index between CS2+ and CS–, $t_{25}$ = 3.269, p=0.003; ***Figure 2e***). Post hoc *t*-test showed that the fear recovery index between CS1+ and CS2+ was significantly different ($t_{25}$=–4.114, p<0.001; ***Figure 2e***), aligning well with previous literature that suggested the cue specificity of the fear reconsolidation effect (***Lee et al., 2017***; ***Monfils et al., 2009***). Similarly, these results remain robust even with the 'non-learners' included in the analysis (***Figure 2—figure supplement 1***).

Altogether, in line with our hypothesis, the return of fear memory measured by the reinstatement effect showed distinct patterns at different timescales. The retrieval-extinction procedure diminished fear SCRs for both the reminded and non-reminded CS+, engendering cue-independent amnesia in the short-term test (30 min). On the other hand, at the long-term (24 hr) fear memory test, fear responses were only diminished for the reminded CS+ (but not the non-reminded CS+), consistent with previous literature on memory reconsolidation. Interestingly, SCR measures of the fear memory re-appeared in the medium-term (6 hr) test, and there was no difference between fear responses elicited by the reminded and non-reminded CS+, suggesting that the 6 hr timescale may be at a 'limbo' stage where both the immediate and the delayed (reconsolidation) memory update mechanisms are not at work.

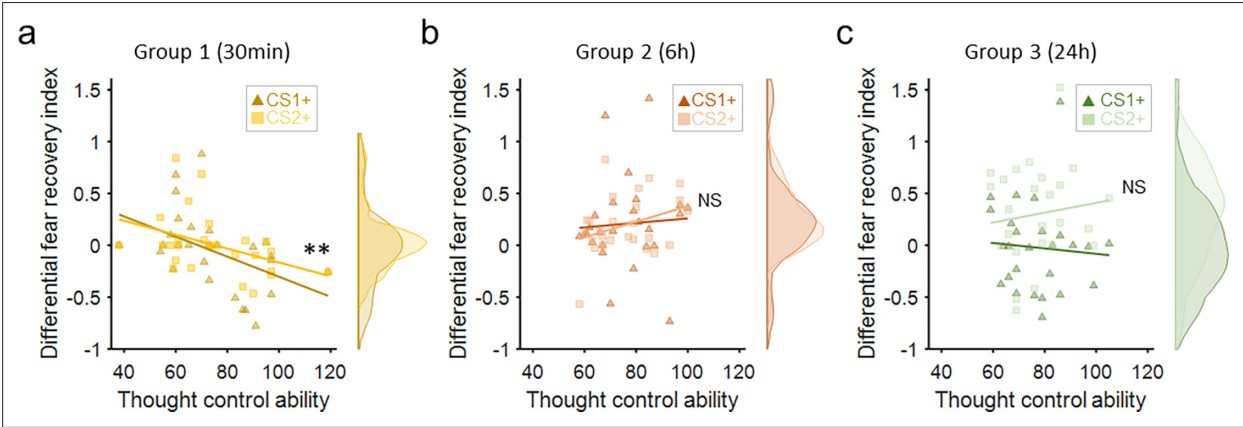

**Figure 3.** Fear recovery as a function of thought-control abilities. (**a**) Thought control ability was significantly correlated with fear recovery index in the 30 min group (p=0.003, Bonferroni correction), but not in the 6 hr (**b**) or 24 hr group (**c**, Ps>0.7). The violin graphs indicate the distribution of fear recovery index across subjects (***Figure 2e***). \*\*p<0.01. NS: nonsignificant.

## Fear amnesia as a function of the thought-control ability

Finally, to assess whether the fear reinstatement effects at different timescales were related to subjects' thought-control ability, we first ran a one-way ANOVA to confirm that subjects among different groups did not differ in their thought-control abilities measured by the TCAQ scores ($F_{2,76}$ = 0.154, p=0.857, $\eta^2$=0.004). We then ran separate linear regressions between the differential fear recovery index (differential fear recovery index between CS+ and CS–) and thought-control abilities (TCAQ scores) in the three groups. This analysis yielded significant negative correlation between TCAQ score and fear recovery index for both CS+ (CS1+ and CS2+) in the 30 min group (*Figure 3a*, main effect of thought-control ability, $t_{51}$=–3.446, p=0.003, Bonferroni correction), indicating that subjects endowed with higher thought-control abilities were less likely to experience the return of fear response. However, no significant correlation was observed in the 6 hr or the 24 hr group (*Figure 3b and c*; Ps>0.4). Post hoc analysis of the 30 min group showed significant correlation between the fear recovery index of both CS1+ and CS2+ with subjects' thought-control abilities (*Figure 3a*, CS1+: *r*=–0.461, p=0.015; CS2+: *r*=–0.413, p=0.032). No significant TCAQ and CS+ interactions were found in the three groups (Ps>0.4).

It is worth noting that correlations between TCAQ scores and fear responses in all three groups were not significant in the late phase of acquisition or the last trial of extinction (all Ps>0.10), indicating that the effect of thought-control ability was specifically reflected in the fear recovery index.

In conclusion, thought-control abilities were associated with fear recovery indices only in the short-term (30 min) test for both the reminded and non-reminded CS+ but not at longer timescales (6 hr and 24 hr). The 30 min group results are consistent with previous research, which suggested that people with better capability to resist intrusive thoughts also performed better in motivated forgetting for both declarative and associative memories (*Küpper et al., 2014*; *Wang et al., 2021*).

## dlPFC-dependent short-term fear memory amnesia

In Study 3 (*Figure 4a*), we examined whether intact activity of the right dlPFC (rdlPFC) was also necessary for short-term fear amnesia, in addition to the memory retrieval cue (reminder). To test this hypothesis, we set up a reminder CS+ (CS1+ vs. CS2+) × reminder group (reminder vs. no-reminder) × cTBS (rdlPFC vs. vertex) × trial number four-way ANOVA against the differential SCRs (CS+ minus CS–) for the acquisition and extinction phases. In the acquisition phase, as expected, there were no significant main effects of reminder ($F_{1,71}$ = 1.291, p=0.260, $\eta^2$=0.018), cTBS ($F_{1,71}$< 0.001, p=0.990, $\eta^2$<0.001), CS+ ($F_{1,71}$ = 1.927, p=0.169, $\eta^2$=0.026) or their interactions (all Ps>0.1; *Figure 4b and c*) but a significant main effect of trial number ($F_{9,639}$ = 8.054, p<0.001, $\eta^2$=0.102), indicating successful learning of CS+ stimuli. Indeed, post-hoc tests confirmed that the SCRs of CS+ were significantly higher than that of the CS– in the late phase (last 5 trials) of acquisition in all four groups (CS1+: $t_{18}$=7.735, p<0.001; CS2+: $t_{18}$=8.546, p<0.001 for the R-PFC group; CS1+: $t_{17}$=8.568, p<0.001; CS2+: $t_{17}$=8.478, p<0.001 for the R-VER group; CS1+: $t_{17}$=6.032, p<0.001; CS2+: $t_{17}$=7.722, p<0.001 for the NR-PFC group and CS1+: $t_{19}$=9.110, p<0.001; CS2+: $t_{19}$=6.538, p<0.001 for the NR-VER group, *Figure 4c*).

A similar mixed-effect four-way ANOVA was also performed on the SCRs in the extinction phase, which showed significant effects of trial ($F_{10,710}$ = 8.567, p<0.001, $\eta^2$=0.108), cTBS ($F_{1,71}$ = 6.269, p=0.015, $\eta^2$=0.081), and cTBS×trial interaction ($F_{10,710}$ = 2.292, p=0.030, $\eta^2$=0.031), but no effect of reminder ($F_{1,71}$ = 1.894, p=0.173, $\eta^2$=0.026), CS+ ($F_{1,71}$ = 0.398, p=0.530, $\eta^2$=0.006), or other interactions (all Ps>0.1; *Figure 4b*). To test whether participants across all groups achieved similar levels of extinction, a mixed-effect three-way ANOVA was performed on the last trial of extinction training, and there was no significant effect of reminder (reminder and no-reminder; $F_{1,71}$ = 3.112, p=0.082, $\eta^2$=0.042), cTBS ($F_{1,71}$ = 0.929, p=0.338, $\eta^2$=0.013), CS+ (CS1+ vs. CS2+; $F_{1,71}$ = 1.623, p=0.207, $\eta^2$=0.022) or their interactions (all Ps>0.1; *Figure 4d*). Furthermore, post hoc t-tests showed that there was no difference between CS+ and CS– responses in all groups on the last trial of extinction (CS1+: $t_{18}$=1.310, p=0.207; CS2+: $t_{18}$=1.298, p=0.211 for the R-PFC group; CS1+: $t_{17}$=0.839, p=0.413; CS2+: $t_{17}$=1.090, p=0.291 for the R-VER group; CS1+: $t_{17}$=0.380, p=0.709; CS2+: $t_{17}$=0.624, p=0.541 for the NR-PFC group and CS1+: $t_{19}$=–1.434, p=0.168; CS2+: $t_{19}$=–0.535, p=0.599 for the NR-VER group; *Figure 4d*).

To assess the effects of the right dlPFC on the memory retrieval-related short-term amnesia, we conducted a mixed-effect three-way ANOVA with the within-subject factors CS+ (CS1+ vs. CS2+), between-subject factors reminder (reminder vs. no-reminder), and cTBS (dlPFC vs. vertex) on fear

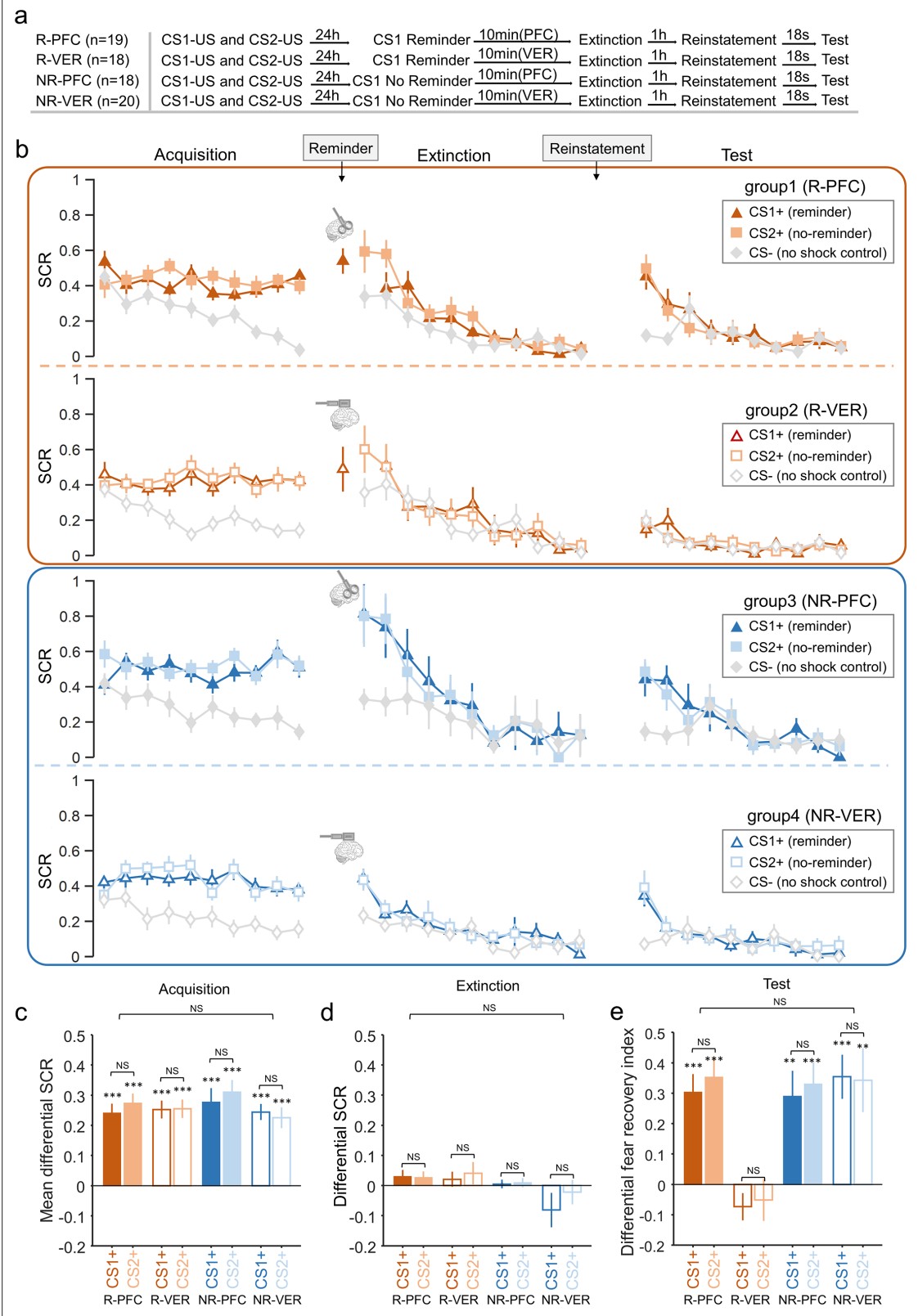

**Figure 4.** Skin conductance response (SCR) responses to conditioned stimuli in the continuous theta-burst stimulation (cTBS) study (Study 3), indicating that both intact dorsolateral prefrontal cortex (dlPFC) and cue reminder are required for short-term memory amnesia. (**a**) Experimental design and timeline. (**b**) SCRs of fear-conditioned stimuli CS1+ (reminder) and CS2+ (no-reminder), and the control stimulus (CS–) across the fear acquisition, extinction, and test phases for each group (R-PFC, R-VER, NR-PFC, and NR-VER groups). (**c**) Mean differential SCRs (CS1+ minus CS– and CS2+ minus

*Figure 4 continued on next page*

*Figure 4 continued*

CS–) in the acquisition phase (trials in the latter half). (**d**) Differential SCRs in the extinction phase (last trial). (**e**) Differential fear recovery index between CS+ and CS– in the test phase. ***p<0.001. *p<0.05. NS: nonsignificant. ANOVA and post-hoc t-test, error bars represent standard errors (n = 19, 18, 18 & 20 for the P-PRC, R-VER, NR-PFC and NR-VER groups, respectively).

recovery index (the difference between the first test trial and the last extinction trial for CS1+ and CS2+), which revealed a significant effect of reminder ($F_{1,71}$ = 8.920, p=0.004, $\eta^2$=0.112), cTBS ($F_{1,71}$ = 7.180, p=0.009, $\eta^2$=0.092), and reminder×cTBS interaction ($F_{1,71}$ = 10.613, p=0.002, $\eta^2$=0.130), but no effect of CS+ ($F_{1,71}$ = 0.518, p=0.474, $\eta^2$=0.007) or other interactions (all Ps>0.1; *Figure 4c*). Post hoc *t*-tests showed that there were significant fear reinstatement responses in the R-PFC group (CS1+: $t_{18}$=5.182, p<0.001; CS2+: $t_{18}$=5.258, p<0.001), the NR-PFC group (CS1+: $t_{17}$=3.496, p=0.003; CS2+: $t_{17}$=4.628, p<0.001) and the NR-VER group (CS1+: $t_{19}$=4.885, p<0.001; CS2+: $t_{19}$=3.262, p=0.004), but not in the R-VER group (CS1+: $t_{17}$=–1.640, p=0.119; CS2+: $t_{17}$=–0.744, p=0.467) (*Figure 4e*), suggesting that both memory retrieval and intact dlPFC function are necessary for the short-term fear amnesia.

Finally, we went on to examine whether the fear reinstatement response was related to the thought-control ability. A two-way ANOVA of reminder (reminder vs. no-reminder) × cTBS (rdlPFC vs. vertex) confirmed that there were no significant effects of reminder ($F_{1,71}$ = 0.366, p=0.547, $\eta^2$=0.005), cTBS ($F_{1,71}$ = 0.606, p=0.439, $\eta^2$=0.008), or their interaction ($F_{1,71}$ = 1.216, p=0.274, $\eta^2$=0.017) on the TCAQ scores. We then conducted simple linear regressions on different groups (as in Study 2) and found a significant correlation between the thought-control ability and differential fear recovery index in the R-VER group (*Figure 5b*, $t_{33}$=–3.283, p=0.008, Bonferroni correction across four groups), consistent with the results of the short-term (30 min) amnesia effect in Study 2 (*Figure 3a*). Further post hoc analysis of the individual CS+ showed that the correlations between thought-control ability and differential fear recovery index were significant for both CS+ (CS1+: r=–0.557, p=0.016; CS2+: r=–0.483, p=0.042). However, there was no such correlation in the R-PFC group (*Figure 5a*, $t_{35}$=0.719, p=0.477), the NR-PFC group (*Figure 5c*, $t_{33}$=–1.334, p=0.192), or the NR-VER group (*Figure 5d*, $t_{37}$=–0.025, p=0.980), and the interaction effects of thought-control ability and CS+ (CS1+ and CS2+) were not significant in all four groups (Ps>0.3). Importantly, there was no significant correlation between thought-control ability and SCR in the late phase of acquisition or the last trial of extinction in four groups (all Ps>0.10), suggesting a specific relationship between thought-control ability and short-term fear recovery.

## Discussion

Our results provide direct evidence to support the hypothesis that memory reactivation might engage distinct cognitive mechanisms for fear memory modulation, which can be separated by their temporal dynamics, cue specificity, and the dependence of thought-control ability. In line with previous research on fear memory reconsolidation (*Liu et al., 2014*), our research replicated the reported results that the retrieval-extinction paradigm blocked fear responses to the reminded CS+ while leaving non-reminded CS+ response intact (cue-specificity) in the long-term test (*Figure 6a and b*). However, thought-control ability was not related to the long-term fear amnesia (*Figure 6c*). Therefore, findings of the short-term fear amnesia suggest that the reconsolidation framework falls short to accommodate this more immediate effect (*Figure 6a and b*).

Research in both declarative and associative emotional memories suggests that memory retrieval is not a simple act of obtaining a copy of stored information, but provides a critical time window where the old memory is eligible to be integrated with new information (*Gershman et al., 2013b*; *Hupbach et al., 2007*; *Monfils et al., 2009*). Research in the field of declarative memory has shown that the motivated forgetting effect can appear early (10–30 min) after the memory suppression practices such as the influential TNT protocol (*Anderson and Green, 2001*), suggesting that memory retrieval may also facilitate short-term memory update (*Benoit and Anderson, 2012*; *Engen and Anderson, 2018*). Unlike the slow protein synthesis process involved in memory reconsolidation, the motivated forgetting effect was accompanied by heightened dlPFC activity, diminished hippocampal activation, and altered functional connectivity between these brain regions and hence more context adaptive (*Anderson et al., 2004*; *Wimber et al., 2015*).

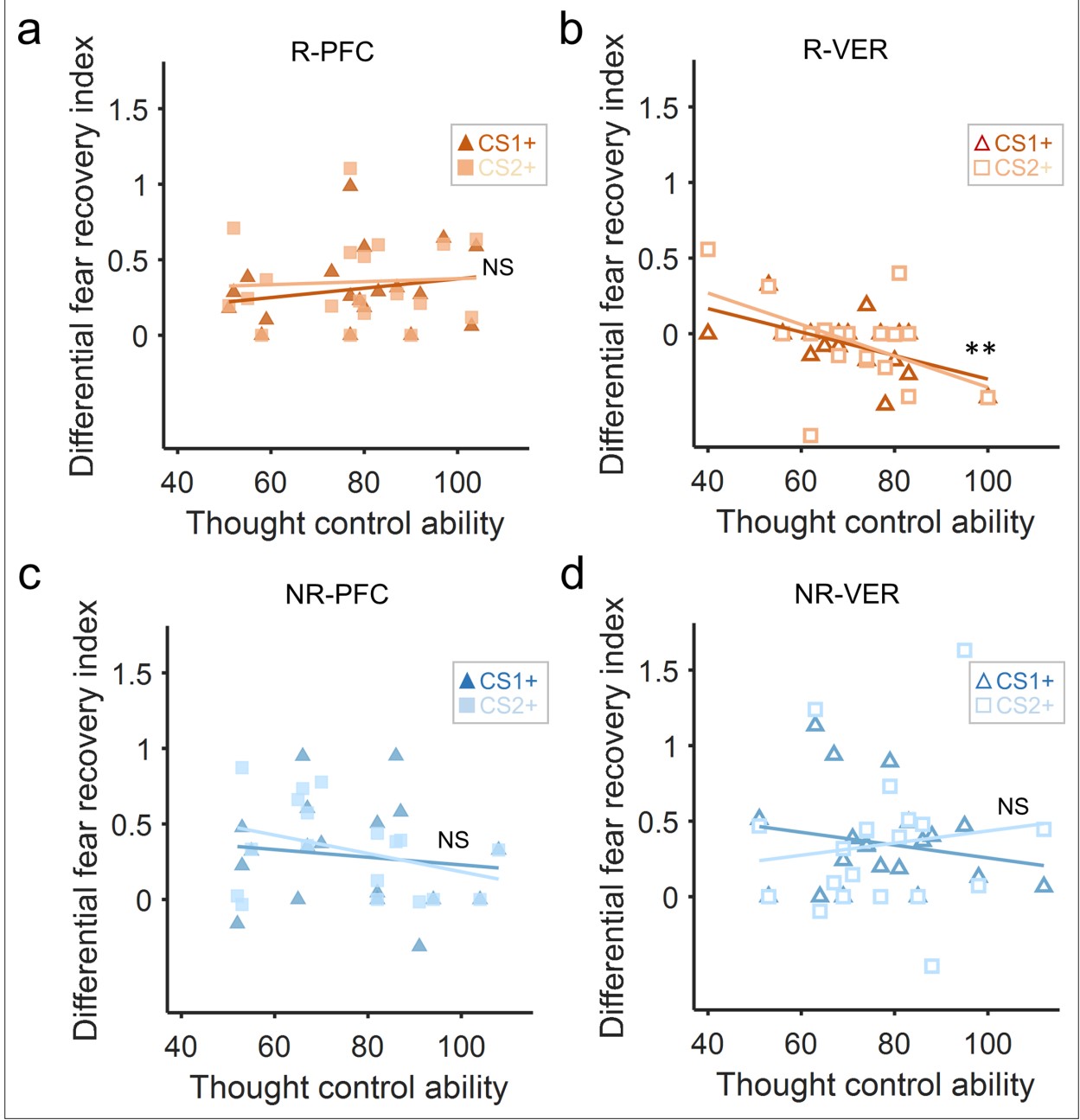

**Figure 5.** The correlation between the differential fear recovery index and thought-control ability in four continuous theta-burst stimulation (cTBS) groups. (**a**) There was no significant correlation between differential fear recovery index and thought-control abilities in the R-PFC group (p>0.4). (**b**) However, in the R-VER group, the correlation between thought-control ability scores and the differential fear recovery index was significant (p=0.008, Bonferroni correction), with high thought-control ability participants showing less fear recovery for both CS+ (CS1+ and CS2+). Such correlation was not observed in the (**c**) NR-PFC or the (**d**) NR-VER group (Ps>0.4). **p<0.01, NS: nonsignificant.

Previous studies indicate that a memory suppression mechanism can be characterized by three distinct features: first, the memory suppression effect tends to emerge early, usually 10–30 min after memory suppression practice and can be transient (*MacLeod and Macrae, 2001*; *Saunders and MacLeod, 2002*); second, the memory suppression practice seems to directly act upon the unwanted memory itself (*Levy and Anderson, 2002*), such that the presentation of other cues originally associated with the unwanted memory also fails in memory recall (cue independence); third, the magnitude of memory suppression effects is associated with individual differences in control abilities over intrusive thoughts (*Küpper et al., 2014*). The short-term fear amnesia effects we identified are fundamentally

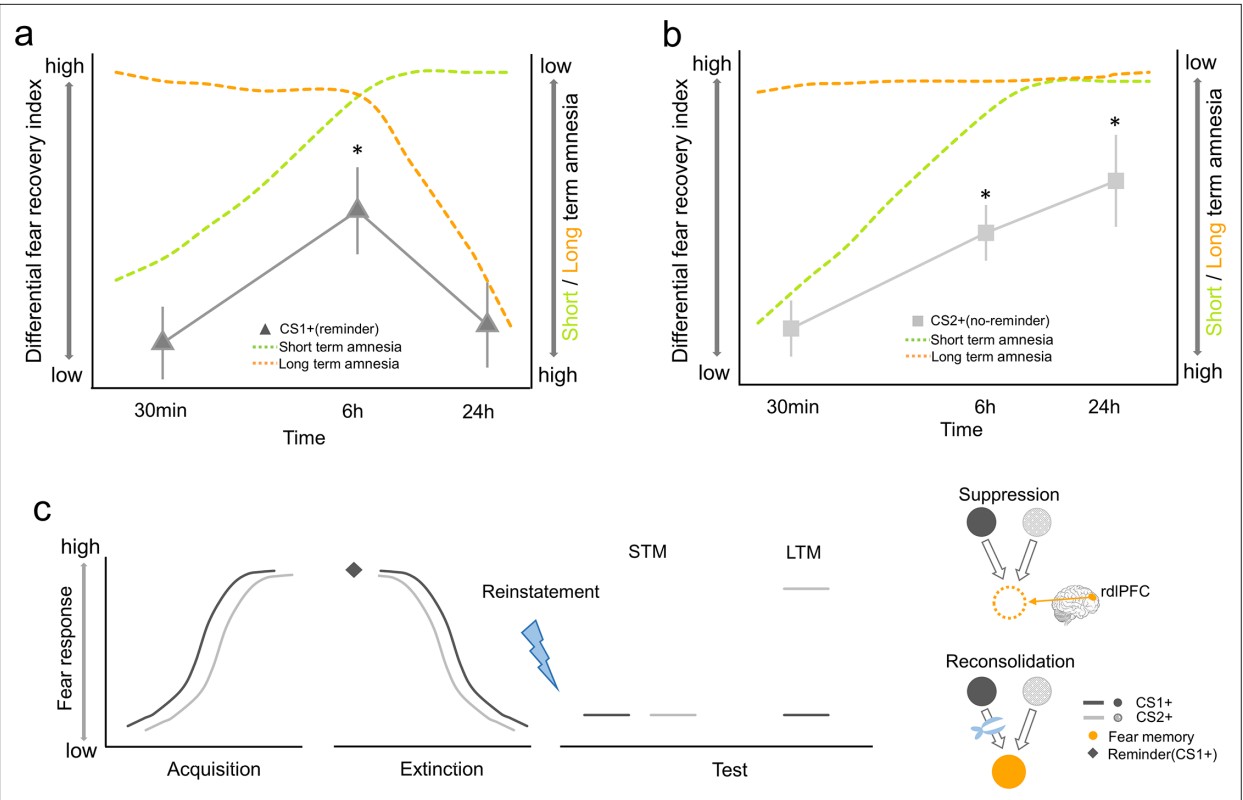

**Figure 6.** Time courses of short- and long-term amnesia. (**a**) At the short interval (30 min to 1 hr), fear recovery of the reminded CS (CS1+) is inhibited (green). As time progresses (from 6 hr to 24 hr), the amnesia effect is likely due to the fear memory reconsolidation effect emerging later (orange). Actual skin conductance response (SCR) data in black solid line. (**b**) Fear amnesia of the non-reminded CS (CS2+) is only evident at the 30 min interval and such effect starts to decay as test interval increases (green). However, long-term amnesia does not generalize to CS2+ (orange, cue dependence). The observed SCR data (gray solid line) paralleled the prediction from both the short-term and the long-term effects as the interval length increased (from 30 min to 24 hr). (**c**) Schematics depicting the effect of cue reminder on fear memory retention. After both CS1+ (black) and CS2+ (gray) successfully elicit fear responses in the acquisition phase, CS1+ is reminded (black) before both CS+ go through the extinction training. The lack of both CS1+ and CS2+ fear responses in the short-term memory (STM) test (30 min) might be explained by the dorsolateral prefrontal cortex (dlPFC) dependent direct suppression effect (dotted circle of US representation). On the other hand, the cue-specific fear amnesia effect in the long-term memory (LTM) test (24 hr) of CS1+ but not CS2+ could be attributed to the reconsolidation effect specific to CS1+. *p<0.05. Error bars represent standard errors.

different from memory reconsolidation effects and fit closely with the characteristics of the memory suppression effect. Inspired by the similarities between our results and suppression-induced declarative memory amnesia (*Gagnepain et al., 2017*), we speculate that the retrieval-extinction procedure might facilitate a spontaneous memory suppression process and thus yield a short-term amnesia effect. Accordingly, the activated fear memory induced by the retrieval cue would be subjected to an automatic fear memory suppression through the extinction training (*Anderson and Floresco, 2022*).

In our experiments, subjects were not explicitly instructed to suppress their fear expression, yet the retrieval-extinction training significantly decreased short-term fear expression. These results are consistent with the short-term amnesia induced with the more explicit suppression intervention (*Anderson et al., 1994*; *Kindt and Soeter, 2018*; *Speer et al., 2021*; *Wang et al., 2021*; *Wells and Davies, 1994*). It is worth noting that although consciously repelling unwanted memory is a standard approach in memory suppression paradigm, it is possible that the engagement of the suppression mechanism can be unconscious. For example, in the RIF paradigm, recall of a stored memory impairs the retention of related target memory, and this forgetting effect emerges as early as 20 min after the retrieval procedure, suggesting memory suppression or inhibition can occur in a more spontaneous and automatic manner (*Imai et al., 2014*). Moreover, subjects with trauma histories exhibited more suppression-induced forgetting for both negative and neutral memories than those with little or no trauma (*Hulbert and Anderson, 2018*). Similarly, people with higher self-reported thought-control capabilities showed more severe cue-independent memory recall deficit, suggesting that suppression

mechanism is associated with individual differences in spontaneous control abilities over intrusive thoughts (*Küpper et al., 2014*). It has also been suggested that similar automatic mechanisms might be involved in organic retrograde amnesia of traumatic childhood memories (*Schacter et al., 2012*; *Schacter et al., 1996*).

Therefore, it is theoretically possible that similar brain mechanisms, such as the dynamic brain activity and connectivity modulation, involved in memory suppression may be deployed spontaneously and act as a general cognitive mechanism to cope with intrusive memories in daily life. Indeed, recently researchers have proposed that the standard retrieval-extinction paradigm to study fear memory might also involve automatic memory suppression (*Anderson and Floresco, 2022*). Consistent with this hypothesis, we showed that the retrieval cue was necessary to produce the short-term amnesia effect (*Figure 1*) and critically, our results fitted well with the three key characteristics of the active suppression mechanism: temporal dynamics, cue specificity, and the dependence of thought-control ability (*Figures 2 and 3*).

To calibrate the temporal boundaries of the short- and long-term effects, we also included a separate group (medium-term group) where fear expression was tested 6 hr after the retrieval-extinction training. Interestingly, fear returned at this time point (*Figure 6a and b*), confirming the 6 hr lower time boundary for the hypothesized memory reconsolidation in previous animal and human studies (*Agren, 2014*; *Agren et al., 2012*; *Debiec et al., 2002*; *Kindt and Soeter, 2018*; *Lee, 2008*; *Nader et al., 2000*; *Speer et al., 2021*). However, the return of fear expression in the medium-term group also indicates that the short-term fear amnesia is transient, laying out the upper time boundary for the hypothesized fear memory suppression effect. Although mixed results have been reported regarding the durability of suppression effects in the declarative memory studies (*Meier et al., 2011*; *Storm et al., 2012*), future research will be needed to investigate whether the short-term effect we observed is specifically related to associative memory or the spontaneous nature of suppression as in RIF (*Figure 6c*). Previous research also showed that separate mnemonic processes can be involved after emotional memory retrieval. For example, by varying the duration length and repetition number of memory cue re-exposure, different levels of memory persistence (reconsolidation, extinction, or a 'limbo' state in between) can be induced, suggesting a neural titration mechanism for memory de-stabilization and re-stabilization (*Faliagkas et al., 2018*; *Merlo et al., 2014*). These works identified important 'boundary conditions' of memory retrieval in affecting the retention of the maladaptive emotional memories. In our study, however, we showed that even within a boundary condition previously thought to elicit memory reconsolidation, mnemonic processes other than reconsolidation could also be at work, and these processes jointly shape the persistence of fear memory.

Our results also corroborate with a recent study showing that blocking fear reconsolidation using post-retrieval repetitive transcranial magnetic stimulation (rTMS) did not influence the short-term fear memory expression (*Borgomaneri et al., 2020*). However, as we demonstrated in Studies 2 and 3, memory retrieval and intact rdlPFC function were both necessary for the short-term fear amnesia after the extinction training. Therefore, instead of 'intact short-term fear expression by the rTMS over the dlPFC', their results could also be interpreted such that the rTMS on the dlPFC demolished the otherwise short-term fear amnesia induced by the memory retrieval cue. Indeed, a series of neuroimaging studies have revealed that the prefrontal neural network including anterior cingulate cortex, anterior ventrolateral prefrontal cortex, and the dlPFC is directly engaged in purging the unwanted declarative memory from consciousness (*Anderson et al., 2004*; *Benoit and Anderson, 2012*; *Levy and Anderson, 2002*). Moreover, previous research has established a causal relationship between dlPFC activity and the memory suppression mechanism underlying RIF using transcranial direct current stimulation (*Penolazzi et al., 2014*; *Stramaccia et al., 2017*; *Valle et al., 2020*). In our experiment (Studies 2 and 3), when the dlPFC function was intact, subjects' thought-control abilities were positively correlated with fear amnesia in the test phase. Such correlations disappeared in the no-reminder groups (NR-VER and NR-PFC) and the R-PFC group (*Figures 3 and 5*), establishing a causal link between the dlPFC activity and short-term fear amnesia.

It should be noted that while our long-term amnesia results were consistent with the fear memory reconsolidation literatures, there were also studies that failed to observe fear prevention (*Chalkia et al., 2020a*; *Chalkia et al., 2020b*; *Schroyens et al., 2023*). Although the memory reconsolidation framework provides a viable explanation for the long-term amnesia, more evidence is required to validate the presence of reconsolidation, especially at the neurobiological level (*Elsey et al., 2018*).

While it is beyond the scope of the current study to discuss the discrepancies between these studies, one possibility to reconcile these results concerns the procedure for the retrieval-extinction training. It has been shown that the eligibility for old memory to be updated is contingent on whether the old memory and new observations can be inferred to have been generated by the same latent cause (*Gershman et al., 2017*; *Gershman and Niv, 2012*). For example, prevention of the return of fear memory can be achieved through a gradual extinction paradigm, which is thought to reduce the size of prediction errors to inhibit the formation of new latent causes (*Gershman et al., 2013a*). There-fore, the effectiveness of the retrieval-extinction paradigm might depend on the reliability of such a paradigm in inferring the same underlying latent cause. Furthermore, other studies highlighted the importance of memory storage per se and suggested that memory retention was encoded in the memory engram cell ensemble connectivity, whereas the engram cell synaptic plasticity is crucial for memory retrieval (*Ryan et al., 2015*; *Tonegawa et al., 2015a*; *Tonegawa et al., 2015b*). It remains to be tested how the cue-independent short-term and cue-dependent long-term amnesia effects we observed could correspond to the engram cell synaptic plasticity and functional connectivity among engram cell ensembles (*Figure 6*). This is particularly important, since the cue-independent charac-teristic of the short-term amnesia suggests that either different memory cues fail to evoke engram cell activities, or the retrieval-extinction training transiently inhibits connectivity among engram cell ensembles. Finally, SCR is only one aspect of the fear expression, how the retrieval-extinction para-digm might affect subjects' other emotional (such as the startle response) and cognitive fear expres-sions such as reported fear expectancy needs to be tested in future studies since they do not always align with each other (*Kindt et al., 2009*; *Sevenster et al., 2012*; *Sevenster et al., 2013*).

These results help shed light on the dynamics of memory modulation after memory activation and designing novel treatment of psychiatric disorders caused by excessive fear or anxiety (*Chen et al., 2021*). Memory reconsolidation and suppression both have been studied thoroughly in separate fields. Specifically, aversive memories are usually associated with different retrieval cues, and inter-fering memory reconsolidation only blocks fear memory recall from the retrieval memory probe but not the others (*Debiec et al., 2002*; *Kindt and Soeter, 2018*; *Lee, 2008*; *Nader et al., 2000*; *Speer et al., 2021*). On the other hand, memory suppression is cue-independent and directly suppresses the fear memory trace yet only effective with limited durability (*MacLeod and Macrae, 2001*). Although future research is clearly needed to understand the brain mechanisms for the short-term amnesia and its potential connection to the memory suppression effect, especially in relevance to clinical popu-lations such as posttraumatic stress disorder and phobia patients (*Homan et al., 2019*; *Stramaccia et al., 2021*), our results provide a general framework to highlight the potential of memory retrieval to triggering different memory updating mechanisms with unique temporal dynamics.

## Materials and methods

### Participants

All participants were students recruited from Peking University. They were right-handed with normal or corrected to normal vision and had not participated in electric shock-related experiments before. None of the participants had current or history of psychiatric illness, nor any contraindication on the application of transcranial magnetic stimulation (TMS) (*Rossi et al., 2009*; *Rossini et al., 2015*). All participants provided informed consent and were remunerated for their participation. This study was approved by the institutional review board of the school of psychological and cognitive sciences at Peking University.

We first conducted a power analysis (G*Power) to determine the number of participants sufficient to detect a reliable effect (*Faul et al., 2009*). Based on the reported effect size of the reinstated fear memory between the treatment and control groups on fear memories reported in the previous literature ($\eta^2$=0.19, 0.216, 0.24; $N$=40, 55, 30 of sample size, respectively; *Kindt and Soeter, 2018*; *Kindt et al., 2009*; *Wang et al., 2021*), 52 participants for both groups (Study 1: reminder group and no-reminder group) were enough to detect a significant interaction effect ($\alpha$=0.05, $\beta$=0.9, 2 (groups) × 2 (phases) two-way ANOVA interaction effect). Similarly, a total of 72 participants for three groups (Study 2: 30 min group, 6 hr group, and 24 hr group) were required to detect a reliable effect ($\alpha$=0.05, $\beta$=0.9, 3 (groups) × 2 (phases) × 2 (CS+) three-way ANOVA interaction effect) (*Heo and Leon, 2010*). For the cTBS study, based on the reported effect size of reinstated fear memory ($\eta^2$=0.144, 0.19,

0.216, 0.24; *N*=79, 40, 55, 30 total sample size, respectively; *Kindt and Soeter, 2018*; *Kindt et al., 2009*; *Wang et al., 2021*), a total of 72 participants (Study 3) were needed to detect a significant interaction effect ($\alpha$=0.05, $\beta$=0.8, 4 (groups) × 2 (phases)×2 (CS+) three-way ANOVA interaction effect).

In Study 1, we first recruited a total of 75 human subjects (41 females; mean age = 21.3, SD = 2.21). Eight participants discontinued after day 1 due to their non-measurable spontaneous CS signal (the SCR) toward different CSs (non-responders, SCR response to any CS was less than 0.02 µS, *n*=0 and 8; reminder group and no-reminder group, respectively). Ten participants (7 females; mean age = 20.6, SD = 1.79) who were 'non-learners' during fear acquisition (*n*=3 and 1 in both groups, respectively) or extinction (*n*=3 and 3, respectively) phase were also excluded from further analysis. The criteria for 'non-learners' in the fear acquisition include smaller mean SCR of CS+ (CS that was partially paired with the electric shock) than that of the CS− (CS that was never paired with shock) trials in the latter half of acquisition, and a smaller mean SCR difference between CS+ and CS− in the latter than the first half trials of fear acquisition on day 1. Similarly, the criteria for 'non-learners' in the fear extinction are larger mean SCRs of CS+ than the CS− responses in both the last trial and the latter-half trials of extinction and a larger mean SCR difference between CS+ and CS− in the latter than the first half trials during fear extinction on day 2 (see the exclusion criteria in *Supplementary file 1*). Therefore, we had a total of 57 subjects for data analysis in Study 1 (30 for the reminder group, 17 females, mean age = 21.4, SD = 2.37, and 27 for the no-reminder group, 15 females, mean age = 21.62, SD = 2.13). The group×gender interaction effect on participants' age was not statistically significant ($F_{1,49}$ = 0.137, p=0.713, $\eta^2$=0.003, two-way ANOVA), and gender ($\chi^2_1$ = 0.007, p=0.933) and age ($t_{55}$=−0.376, p=0.708) were not significantly different between two groups.

In Study 2, we first recruited 119 participants (56 females; mean age = 20.85, SD = 2.59). Thirty-seven participants discontinued after day 1 since they were 'non-responders' (n=19, 5, and 13; 30 min group, 6 hr group, and 24 hr group, respectively). Additionally, two participants (*n*=1 and 1; 30 min group and 24 hr group, respectively) failed to show the evidence of fear acquisition, and one participant (30 min group) failed to show the evidence of fear extinction on day 2 and was also excluded from further analyses. Finally, we had a total of 79 subjects for data analysis in Study 2 (27 for the 30 min group, 15 females, mean age = 21.4, SD = 2.46; 26 for the 6 hr group, 12 females, mean age = 20.54, SD = 1.74; and 26 for the 24 hr group, 12 females, mean age = 20.96, SD = 2.990). The interaction effect of gender × group on participants' age was not statistically significant ($F_{2,73}$ = 0.213, p=0.809, $\eta^2$=0.006, two-way ANOVA), and gender ($\chi^2_2$=0.021, p=0.990) or age ($F_{2,76}$ = 0.661, p=0.519, $\eta^2$=0.017) was not significantly different across three groups.

In Study 3, we first recruited a total of 90 participants (47 females; mean age = 21.01, SD = 2.44). Fifteen participants (8 females; mean age = 21.2, SD = 1.22) were excluded from further analysis since they failed to show SCR response to any stimulus (CS1+, CS2+, or CS−) (non-responders, *n*=3, 4, 4, 4; R rdlPFC group, R vertex group, NR rdlPFC group, and NR vertex group, respectively, see *Supplementary file 1*; *de Voogd and Phelps, 2020*; *Dunsmoor et al., 2015*; *Dunsmoor et al., 2017*; *Hu et al., 2018*; *Raio et al., 2017*; *Schiller et al., 2013*; *Schiller et al., 2010*; *Schiller et al., 2012*). Our final sample included 75 participants for data analysis (19 for the reminder right dlPFC group, 10 females, mean age = 22.05, SD = 2.44; 18 for the reminder vertex group, 9 females, mean age = 20.67, SD = 2.68; 18 for the no-reminder right dlPFC group, 10 females, mean age = 20.28, SD = 2.85; and 20 for the no-reminder vertex group, 10 females, mean age = 20.85, SD = 2.43).

## CSs and psychophysiological stimulation

For Study 1, we employed two squares with different colors (yellow and blue) that served as CS+ and CS−. During fear acquisition, the CS+ was paired with a mild electrical shock (US) on a 37.5% partial reinforcement scheme (6 CS+ paired with electric shock and 10 CS+ with no shock), and the CS− was never paired with a shock (10 CS−). A pseudorandom CS delivery order was generated for the fear acquisition, extinction, and test phase for two groups with the rule that no same trial type (CS+ and CS−) repeated more than twice. Assignment of square colors to the CS (CS+ and CS−) was counterbalanced across participants.

For Studies 2 and 3, three squares with different colors (yellow, red, and blue) were used as CS1+, CS2+, and CS−, respectively. During the fear acquisition phase, both CS1+ and CS2+ were paired with electric shocks (US) on a 37.5% partial reinforcement scheme, and the CS− was not paired with shocks (10 non-reinforced presentations of CS1+, CS2+, and CS− each, intermixed with an extra of 6 CS1+

and 6 CS2+ that co-terminated with the shock). Again, a pseudorandom stimulus order was generated for fear acquisition and extinction phases of three groups with the rule that no same trial type (CS1+, CS2+, and CS–) repeated more than twice. In the test phase, to exclude the possibility that the difference between CS1+ and CS2+ was simply caused by the presentation sequence of CS1+ and CS2+, half of the participants completed the test phase using a pseudorandom stimuli sequence and the identities of CS1+ and CS2+ reversed in the other half of the participants. For all three groups, the CS colors were counterbalanced across participants.

The US was a mild electrical shock delivered to the right wrist of participants via a DS-5 Isolated Bipolar Constant Current Stimulator (Digitimer, Welwyn Garden City, UK). The US levels were set by the participants themselves, starting with a low shock level (5v) and gradually increased and settled to a level that they described as 'uncomfortable, but not painful' (with the maximum level of 10v). The duration of all electric shocks was 200 ms with 50 pulses per second.

SCR measurements were collected using two Ag-AgCl electrodes attached to the tips of the index and middle fingers of each subject's left hand. All skin conductance data were recorded via the Biopac MP160 BioNomadix System and analyzed using the Acknowledgement 5.0 software. The SCR level for each trial was calculated as the amplitude difference (in micro siemens) from peak to trough during the 0.5–4.5 s window after the CS (CS SCR) or US (US SCR) onset, and responses below 0.02 µS were encoded as zero. The raw SCR scores were divided by each subject's mean US responses and then square-root-transformed to normalize SCR across individuals and groups (*Wang et al., 2021*).

cTBS, a specific form of rTMS, was applied with a Magstim Rapid 2 system (Magstim, Whitland, Wales, UK). Before the experiments, we determined the stimulation intensity of the cTBS protocol that was at 80% of the individual active motor threshold. Each TMS burst consisted of three stimuli pulses at 50 Hz, with each train being repeated every 200 ms (5 Hz). In this cTBS protocol, a 40 s train of uninterrupted TBS is given (600 pluses) (*Borgomaneri et al., 2020*; *Su et al., 2022*). During the stimulation, the cTBS was applied either over the right dlPFC or the vertex, which was determined by standard F4 and Cz locations based on international electroencephalogram 10–20 system measurements (*Censor et al., 2010*).

The control ability over intrusive thoughts was measured by the 25-item TCAQ scale (*Wells and Davies, 1994*). Participants were asked to rate on a five-point Likert-type scale the extent to which they agreed with the statement from 1 (*completely disagree*) to 5 (*completely agree*). At the end of the experiments, all participants completed the TCAQ scale to assess their perceived control abilities over intrusive thoughts in daily life (*Küpper et al., 2014*). The internal consistency (Cronbach's alpha) of the TCAQ scale was 0.92, and the reliability coefficient was satisfactory ($r$=0.88).

To measure the extent to which fear returns after the presentation of US (electric shock) in the test phase, we defined the fear recovery index as the SCR difference between the first test trial and the last extinction trial for a specific CS for each subject. Similarly, in Studies 2 and 3, differential fear recovery index was defined as the difference between fear recovery indices of CS+ and CS– for both CS1+ and CS2+.

## Experimental procedure

Studies 1, 2, and 3 were carried out for 2 or 3 consecutive days with three phases. Before the experiments, all participants gave informed consent. During the experiments, subjects were required to stay relaxed and still, focus on the computer screen, and pay attention to the relationship between color stimuli (CS) and the electric shock (US).

Day 1: Acquisition. For Study 1, two groups of participants underwent a fear-conditioning paradigm (acquisition) where the CS+ was partially (37.5%) paired with the US, and the CS– was never paired with the shock. For Studies 2 and 3, participants underwent fear conditioning using three color squares as the CSs (CS1+, CS2+, and CS–). The CSs were presented for 4 s, each with a 10–12 s variable inter-trial interval (ITI) in all the studies (1, 2, and 3).

Day 2: Retrieval-extinction phase. For Study 1, the fear memory was reactivated before extinction in the reminder group. During reactivation, the CS+ was presented once without the delivery of the US. Followed by a 10 min break, the reminder group underwent extinction training with 10 CS+ and 11 CS– presentation non-reinforced. In the no-reminder group, the fear memory was not reactivated (no CS+ reminder). To match with the timeline of the reminder group, the no-reminder group subjects still needed to wait for 10 min after the beginning of the experiment on day 2 and then underwent

extinction training with 11 CS+ and 11 CS+ non-reinforced presentation. During the 10 min break, all participants were asked to rest and stay still.

For Study 2, only one of the CS+ (CS1+) was presented once without the US during reactivation in all three groups. After a 10 min break, all three groups underwent extinction training with 10 CS1+, 11 CS2+, and 11 CS– non-reinforced presentation.

For Study 3, for the reminder groups (reminder-dlPFC and reminder-vertex groups), only one of the CS+ (CS1+) was presented once (without the US delivery) as the retrieval trial. No CS+ reminder was delivered for the no-reminder groups (no-reminder dlPFC and no-reminder vertex groups). Additionally, during the 10 min break (8 min after the supposed CS+ reminder delivery), cTBS was applied either over the right dlPFC (dlPFC groups: PFC) or the vertex (vertex groups: VER), resulting in a 2 (reminder vs. no-reminder) × 2 (dlPFC vs. vertex) of 4 different groups (reminder-dlPFC: R-PFC, reminder-vertex: R-VER, no-reminder dlPFC: NR-PFC, no-reminder vertex: NR-VER).

Day 2 or day 3: Reinstatement and test phases. For Study 1, 30 min after the extinction training, both groups received four un-signaled electric shocks with 10 s to 12 s ITI (reinstatement). Eighteen seconds after the reinstatement phase, all subjects were presented with CS+ and CS– 10 times each without electric shocks and their SCRs were recorded (test phase).

For Study 2, participants were assigned into three groups: the 30 min group, the 6 hr group, and the 24 hr group. All participants received four un-signaled electric shocks with 10 s to 12 s ITI (reinstatement) after their corresponding time delays (30 min, 6 hr, or 24 hr) between extinction and reinstatement. Eighteen seconds after the reinstatement phase, all subjects were tested via the presentation of CS1+, CS2+, and CS– 10 times each without electric shocks associated.

For Study 3, the reinstatement and testing procedures were the same as in Study 2 except that they were conducted 1 hr after the extinction training.

## Acknowledgements

This work was supported by the National Science and Technology Innovation 2030 Major Program (2021ZD0203702), National Natural Science Foundation of China Grants (32441111, 32071090) to JL, National Institute of Health, USA (R01MH122611, R01MH123069) to DS.

## Additional information

### Funding

| Funder | Grant reference number | Author |
|---|---|---|
| National Science and Technology Major Project | 2021ZD0203702 | Jian Li |
| National Natural Science Foundation of China | 32441111 | Jian Li |
| National Institute of Mental Health | R01MH122611 | Daniela Schiller |
| National Institute of Mental Health | R01MH123069 | Daniela Schiller |
| National Natural Science Foundation of China | 32071090 | Jian Li |

The funders had no role in study design, data collection and interpretation, or the decision to submit the work for publication.

### Author contributions

Yinmei Ni, Data curation, Formal analysis, Visualization, Methodology, Writing – review and editing; Ye Wang, Conceptualization, Formal analysis, Investigation, Visualization, Methodology, Writing – original draft; Zijian Zhu, Conceptualization; Jingchu Hu, Formal analysis, Methodology; Daniela Schiller, Conceptualization, Methodology, Writing – original draft, Writing – review and editing; Jian

Li, Conceptualization, Resources, Supervision, Visualization, Methodology, Writing – original draft, Project administration, Writing – review and editing

## Author ORCIDs
Yinmei Ni https://orcid.org/0000-0003-3614-1828
Daniela Schiller https://orcid.org/0000-0002-0357-7724
Jian Li https://orcid.org/0000-0002-3941-2622

## Ethics
Human subjects: All participants provided informed consent and were remunerated for their participation. This study was approved by the institutional review board of school of psychological and cognitive sciences at Peking University.

Reviewer #1 (Public review): https://doi.org/10.7554/eLife.98652.4.sa1
Reviewer #2 (Public review): https://doi.org/10.7554/eLife.98652.4.sa2
Author response https://doi.org/10.7554/eLife.98652.4.sa3

---

# Additional files

## Supplementary files
Supplementary file 1. Participant exclusion criteria.

MDAR checklist

## Data availability
The data and analysis codes for all studies are publicly accessible at https://osf.io/9agvk/.

The following dataset was generated:

| Author(s) | Year | Dataset title | Dataset URL | Database and Identifier |
| --- | --- | --- | --- | --- |
| Ni Y | 2025 | Fear Memory Retrieval | https://osf.io/9agvk/ | Open Science Framework, 9agvk |

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
