## [Editor Report · eLife Assessment]

This manuscript presents **valuable** findings which reveal an intricate pattern of memory expression following retrieval extinction at different intervals from retrieval-extinction to test. The novel advance is in the demonstration that, relative to a standard extinction procedure, the retrieval-extinction procedure more effectively suppresses responses to a conditioned threat stimulus when testing occurs just minutes after extinction. While the data provide **solid** evidence that the "short-term" suppression of responding involves engagement of the dorsolateral prefrontal cortex, there are inconsistencies in the analyses reported which obscure the interpretation and leave some of the claims with limited evidence.

---

## [Referee Report · Reviewer #1 (Public review)]

Summary:

The novel advance by Wang et al is in the demonstration that, relative to a standard extinction procedure, the retrieval-extinction procedure more effectively suppresses responses to a conditioned threat stimulus when testing occurs just minutes after extinction. The authors provide solid evidence to show that this "short-term" suppression of responding involves engagement of the dorsolateral prefrontal cortex.

Strengths:

Overall, the study is well-designed and the results are valuable. There are, however, a few issues in the way that it is introduced and discussed. It would have been useful if the authors could have more explicitly related the results to a theory - it would help the reader understand why the results should have come out the way that they did. More specific comments are presented below.

Please note: The authors appear to have responded to my original review twice. It is not clear that they observed the public review that I edited after the first round of revisions. As part of these edits, I removed the entire section titled Clarifications, Elaborations and Edits

Theory and Interpretation of Results

(1) It is difficult to appreciate why the first trial of extinction in a standard protocol does NOT produce the retrieval-extinction effect. This applies to the present study as well as others that have purported to show a retrieval-extinction effect. The importance of this point comes through at several places in the paper. E.g., the two groups in study 1 experienced a different interval between the first and second CS extinction trials; and the results varied with this interval: a longer interval (10 min) ultimately resulted in less reinstatement of fear than a shorter interval. Even if the different pattern of results in these two groups was shown/known to imply two different processes, there is nothing in the present study that addresses what those processes might be. That is, while the authors talk about mechanisms of memory updating, there is little in the present study that permits any clear statement about mechanisms of memory. The references to a "short-term memory update" process do not help the reader to understand what is happening in the protocol.

In reply to this point, the authors cite evidence to suggest that "an isolated presentation of the CS+ seems to be important in preventing the return of fear expression." They then note the following: "It has also been suggested that only when the old memory and new experience (through extinction) can be inferred to have been generated from the same underlying latent cause, the old memory can be successfully modified (Gershman et al., 2017). On the other hand, if the new experiences are believed to be generated by a different latent cause, then the old memory is less likely to be subject to modification. Therefore, the way the 1st and 2nd CS are temporally organized (retrieval-extinction or standard extinction) might affect how the latent cause is inferred and lead to different levels of fear expression from a theoretical perspective." This merely begs the question: why might an isolated presentation of the CS+ result in the subsequent extinction experiences being allocated to the same memory state as the initial conditioning experiences?

This is not addressed in the paper. The study was not designed to address this question; and that the question did not need to be addressed for the set of results to be interesting. However, understanding how and why the retrieval-extinction protocol produces the effects that it does in the long-term test of fear expression would greatly inform our understanding of how and why the retrieval-extinction protocol has the effects that it does in the short-term tests of fear expression. To be clear; the results of the present study are very interesting - there is no denying that. I am not asking the authors to change anything in response to this point. It simply stands as a comment on the work that has been done in this paper and the area of research more generally.

(2) The discussion of memory suppression is potentially interesting but raises many questions. That is, memory suppression is invoked to explain a particular pattern of results but I, as the reader, have no sense of why a fear memory would be better suppressed shortly after the retrieval-extinction protocol compared to the standard extinction protocol; and why this suppression is NOT specific to the cue that had been subjected to the retrieval-extinction protocol. I accept that the present study was not intended to examine aspects of memory suppression, and that it is a hypothesis proposed to explain the results collected in this study. I am not asking the authors to change anything in response to this point. Again, it simply stands as a comment on the work that has been done in this paper.

(3) The authors have inserted the following text in the revised manuscript: "It should be noted that while our long-term amnesia results were consistent with the fear memory reconsolidation literatures, there were also studies that failed to observe fear prevention (Chalkia, Schroyens, et al., 2020; Chalkia, Van Oudenhove, et al., 2020; Schroyens et al., 2023). Although the memory reconsolidation framework provides a viable explanation for the long-term amnesia, more evidence is required to validate the presence of reconsolidation, especially at the neurobiological level (Elsey et al., 2018). While it is beyond the scope of the current study to discuss the discrepancies between these studies, one possibility to reconcile these results concerns the procedure for the retrieval-extinction training. It has been shown that the eligibility for old memory to be updated is contingent on whether the old memory and new observations can be inferred to have been generated by the same latent cause (Gershman et al., 2017; Gershman and Niv, 2012). For example, prevention of the return of fear memory can be achieved through gradual extinction paradigm, which is thought to reduce the size of prediction errors to inhibit the formation of new latent causes (Gershman, Jones, et al., 2013). Therefore, the effectiveness of the retrieval-extinction paradigm might depend on the reliability of such paradigm in inferring the same underlying latent cause." ***It is perfectly fine to state that "the effectiveness of the retrieval-extinction paradigm might depend on the reliability of such paradigm in inferring the same underlying latent cause..." This is not uninteresting; but it also isn't saying much. Ideally, the authors would have included some statement about factors that are likely to determine whether one is or isn't likely to see a retrieval-extinction effect, grounded in terms of the latent state theories that have been invoked here. Presumably, the retrieval-extinction protocol has variable effects because of procedural differences that affect whether subjects infer the same underlying latent cause when shifted into extinction. Surely, the clinical implications of any findings are seriously curtailed unless one understands when a protocol is likely to produce an effect; and why the effect occurs at all? This question is rhetorical. I am not asking the authors to change anything in response to this point. Again, it stands as a comment on the work that has been done in this paper; and remains a comment after insertion of the new text, which is acknowledged and appreciated.

(4) The authors find different patterns of responses to CS1 and CS2 when they were tested 30 min after extinction versus 24 h after extinction. On this basis, they infer distinct memory update mechanisms. However, I still can't quite see why the different patterns of responses at these two time points after extinction need to be taken to infer different memory update mechanisms. That is, the different patterns of responses at the two time points could be indicative of the same "memory update mechanism" in the sense that the retrieval-extinction procedure induces a short-term memory suppression that serves as the basis for the longer-term memory suppression (i.e., the reconsolidation effect). My pushback on this point is based on the notion of what constitutes a memory update mechanism; and is motivated by what I take to be a rather loose use of language/terminology in the reconsolidation literature and this paper specifically (for examples, see the title of the paper and line 2 of the abstract).

To be clear: I accept the authors' reply that "The focus of the current manuscript is to demonstrate that the retrieval-extinction paradigm can also facilitate a short-term fear memory deficit measured by SCR". However, I disagree with the claim that any short-term fear memory deficit must be indicative of "update mechanisms other than reconsolidation", which appears on Line 27 in the abstract and very much indicates the spirit of the paper. To make the point: the present study has examined the effectiveness of a retrieval-extinction procedure in suppressing fear responses 30 min, 6 hours and 24 hours after extinction. There are differences across the time points in terms of the level of suppression, its cue specificity, and its sensitivity to manipulation of activity in the dlPFC. This is perfectly interesting when not loaded with additional baggage re separable mechanisms of memory updating at the short and long time points: there is simply no evidence in this study or anywhere else that the short-term deficit in suppression of fear responses has anything whatsoever to do with memory updating. It can be exactly what is implied by the description: a short-term deficit in the suppression of fear responses. Again, this stands as a comment on the work that has been done; and remains a comment for the revised paper.

(5) It is not clear why thought control ability ought to relate to any aspect of the suppression that was evident in the 30 min tests - that is, I accept the correlation between thought control ability and performance in the 30 min tests but would have liked to know why this was looked at in the first place and what, if anything, it means. The issue at hand is that, as best as I can tell, there is no theory to which the result from the short- and long-term tests can be related. The attempts to fill this gap with reference to phenomena like retrieval-induced forgetting are appreciated but raise more questions than answers. This is especially clear in the discussion, where it is acknowledged/stated: "Inspired by the similarities between our results and suppression-induced declarative memory amnesia (Gagnepain et al., 2017), we speculate that the retrieval-extinction procedure might facilitate a spontaneous memory suppression process and thus yield a short-term amnesia effect. Accordingly, the activated fear memory induced by the retrieval cue would be subjected to an automatic fear memory suppression through the extinction training (Anderson and Floresco, 2022)." There is nothing in the subsequent discussion to say why this should have been the case other than the similarity between results obtained in the present study and those in the literature on retrieval induced forgetting, where the nature of the testing is quite different. Again, this is simply a comment on the work that has been done - no change is required for the revised paper.

---

## [Referee Report · Reviewer #2 (Public review)]

Summary

The study investigated whether memory retrieval followed soon by extinction training results in a short-term memory deficit when tested - with a reinstatement test that results in recovery from extinction - soon after extinction training. Experiment 1 documents this phenomenon using a between-subjects design. Experiment 2 used a within-subject control and sees that the effect is also observed in a control condition. In addition, it also revealed that if testing is conducted 6 hours after extinction, there is not effect of retrieval prior to extinction as there is recovery from extinction independently of retrieval prior to extinction. A third Group also revealed that retrieval followed by extinction attenuates reinstatement when the test is conducted 24 hours later, consistent with previous literature. Finally, Experiment 3 used continuous theta-burst stimulation of the dorsolateral prefrontal cortex and assessed whether inhibition of that region (vs a control region) reversed the short-term effect revealed in Experiments 1 and 2. The results of control groups in Experiment 3 replicated the previous findings (short-term effect), and the experimental group revealed that these can be reversed by inhibition of the dorsolateral prefrontal cortex.

Strengths

The work is performed using standard procedures (fear conditioning and continuous theta-burst stimulation) and there is some justification of the sample sizes. The results replicate previous findings - some of which have been difficult to replicate and this needs to be acknowledged - and suggest that the effect can also be observed in a short-term reinstatement test.

The study establishes links between the memory reconsolidation and retrieval-induced forgetting (or memory suppression) literatures. The explanations that have been developed for these are distinct and the current results integrate these, by revealing that the DLPFC activity involved in retrieval-extinction short-term effect. There is thus some novelty in the present results, but numerous questions remain unaddressed.

Weakness

The fear acquisition data is converted to a differential fear SCR and this is what is analysed (early vs late). However, the figure shows the raw SCR values for CS+ and CS- and therefore it is unclear whether acquisition was successful (despite there being an "early" vs "late" effect - no descriptives are provided).

In Experiment 1 (Test results) it is unclear whether the main conclusion stems from a comparison of the test data relative to the last extinction trial ("we defined the fear recovery index as the SCR difference between the first test trial and the last extinction trial for a specific CS") or the difference relative to the CS- ("differential fear recovery index between CS+ and CS-"). It would help the reader assess the data if Fig 1e presents all the indexes (both CS+ and CS-). In addition, there is one sentence which I could not understand "there is no statistical difference between the differential fear recovery indexes between CS+ in the reminder and no reminder groups (P=0.048)". The p value suggests that there is a difference, yet it is not clear what is being compared here. Critically, any index taken as a difference relative to the CS- can indicate recovery of fear to the CS+ or absence of discrimination relative to the CS-, so ideally the authors would want to directly compare responses to the CS+ in the reminder and no-reminder groups. In the absence of such comparison, little can be concluded, in particular if SCR CS- data is different between groups. The latter issue is particularly relevant in Experiment 2, in which the CS- seems to vary between groups during the test and this can obscure the interpretation of the result.

In experiment 1, the findings suggest that there is a benefit of retrieval followed by extinction in a short-term reinstatement test. In Experiment 2, the same effect is observed to a cue which did not undergo retrieval before extinction (CS2+), a result that is interpreted as resulting from cue-independence, rather than a failure to replicate in a within-subjects design the observations of Experiment 1 (between-subjects). Although retrieval-induced forgetting is cue-independent (the effect on items that are supressed [Rp-] can be observed with an independent probe), it is not clear that the current findings are similar, and thus that the strong parallels made are not warranted. Here, both cues have been extinguished and therefore been equally exposed during the critical stage.

The findings in Experiment 2 suggest that the amnesia reported in experiment 1 is transient, in that no effect is observed when the test is delayed by 6 hours. The phenomena whereby reactivated memories transition to extinguished memories as a function of the amount of exposure (or number of trials) is completely different from the phenomena observed here. In the former, the manipulation has to do with the number of trials (or total amount of time) that the cues are exposed. In the current Experiment 2, the authors did not manipulate the number of trials but instead the retention interval between extinction and test. The finding reported here is closer to a "Kamin effect", that is the forgetting of learned information which is observed with intervals of intermediate length (Baum, 1968). Because the Kamin effect has been inferred to result from retrieval failure, it is unclear how this can be explained here. There needs to be much more clarity on the explanations to substantiate the conclusions.

There are many results (Ryan et al., 2015) that challenge the framework that the authors base their predictions on (consolidation and reconsolidation theory), therefore these need to be acknowledged. These studies showed that memory can be expressed in the absence of the biological machinery thought to be needed for memory performance. The authors should be careful about statements such as "eliminate fear memores" for which there is little evidence.

The parallels between the current findings and the memory suppression literature are speculated in the general discussion, and there is the conclusion that "the retrieval-extinction procedure might facilitate a spontaneous memory suppression process". Because one of the basic tenets of the memory suppression literature is that it reflects an "active suppression" process, there is no reason to believe that in the current paradigm the same phenomenon is in place, but instead it is "automatic". In other words, the conclusions make strong parallels with the memory suppression (and cognitive control) literature, yet the phenomena that they observed is thought to be passive (or spontaneous/automatic). Ultimately, it is unclear why 10 mins between the reminder and extinction learning will "automatically" supress fear memories. Further down in the discussion it is argued that "For example, in the well-known retrieval-induced forgetting (RIF) phenomenon, the recall of a stored memory can impair the retention of related long-term memory and this forgetting effect emerges as early as 20 minutes after the retrieval procedure, suggesting memory suppression or inhibition can occur in a more spontaneous and automatic manner". I did not follow with the time delay between manipulation and test (20 mins) would speak about whether the process is controlled or automatic. In addition, the links with the "latent cause" theoretical framework are weak if any. There is little reason to believe that one extinction trial, separated by 10 mins from the rest of extinction trials, may lead participants to learn that extinction and acquisition have been generated by the same latent cause.

Among the many conclusions, one is that the current study uncovers the "mechanism" underlying the short-term effects of retrieval-extinction. There is little in the current report that uncovers the mechanism, even in the most psychological sense of the mechanism, so this needs to be clarified. The same applies to the use of "adaptive".

Whilst I could access the data in the OFS site, I could not make sense of the Matlab files as there is no signposting indicating what data is being shown in the files. Thus, as it stands, there is no way of independently replicating the analyses reported.

The supplemental material shows figures with all participants, but only some statistical analyses are provided, and sometimes these are different from those reported in the main manuscript. For example, the test data in Experiment 1 is analysed with a two-way ANOVA with main effects of group (reminder vs no-reminder) and time (last trial of extinction vs first trial of test) in the main report. The analyses with all participants in the sup mat used a mixed two-way ANOVA with group (reminder vs no reminder) and CS (CS+ vs CS-). This makes it difficult to assess the robustness of the results when including all participants. In addition, in the supplementary materials there are no figures and analyses for Experiment 3.

One of the overarching conclusions is that the "mechanisms" underlying reconsolidation (long term) and memory suppression (short term) phenomena are distinct, but memory suppression phenomena can also be observed after a 7-day retention interval (Storm et al., 2012), which then questions the conclusions achieved by the current study.

References:

Baum, M. (1968). Reversal learning of an avoidance response and the Kamin effect. Journal of Comparative and Physiological Psychology, 66(2), 495.

Chalkia, A., Schroyens, N., Leng, L., Vanhasbroeck, N., Zenses, A. K., Van Oudenhove, L., & Beckers, T. (2020). No persistent attenuation of fear memories in humans: A registered replication of the reactivation-extinction effect. Cortex, 129, 496-509.

Ryan, T. J., Roy, D. S., Pignatelli, M., Arons, A., & Tonegawa, S. (2015). Engram cells retain memory under retrograde amnesia. Science, 348(6238), 1007-1013.

Storm, B. C., Bjork, E. L., & Bjork, R. A. (2012). On the durability of retrieval-induced forgetting. Journal of Cognitive Psychology, 24(5), 617-629.

Comments on revisions:

Thanks to the authors for trying to address my concerns.

(1 and 2) My point about evidence for learning relates to the fact that in none of the experiments an increase in SCR to the CSs+ is observed during training (in Experiment 1 CS+/CS- differences are even present from the outset), instead what happens is that participants learn to discriminate between the CS+ and CS- and decrease their SCR responding to the safe CS-. This begs the question as to what is being learned, given that the assumption is that the retrieval-extinction treatment is concerned with the excitatory memory (CS+) rather than the CS+/CS- discrimination. For example, Figures 6A and 6B have short/Long term amnesia in the right axes, but it is unclear from the data what memory is being targeted. In Figure 6C, the right panels depicting Suppression and Reconsolidation mechanisms suggest that it is the CS+ memory that is being targeted. Because the dependent measure (differential SCR) captures how well the discrimination was learned (this point relates to point 2 which the authors now acknowledge that there are differences between groups in responding to the CS-), then I struggle to see how the data supports these CS+ conclusions. The fact that influential papers have used this dependent measure (i.e., differential SCR) does not undermine the point that differences between groups at test are driven by differences in responding to the CS-.

(3, 4 and 5) The authors have qualified some of the statements, yet I fail to see some of these parallels. Much of the discussion is speculative and ultimately left for future research to address.

(6) I can now make more sense of the publicly available data, although the files would benefit from an additional column that distinguishes between participants that were included in the final analyses (passed the multiple criteria = 1) and those who did not (did not pass the criteria = 0). Otherwise, anyone who wants to replicate these analyses needs to decipher the multiple inclusion criteria and apply it to the dataset.

---

## [Author Response]

The following is the authors’ response to the previous reviews

**Reviewer #1 (Public review):**
Introduction & Theory(1) It is difficult to appreciate why the first trial of extinction in a standard protocol does NOT produce the retrieval-extinction effect. This applies to the present study as well as others that have purported to show a retrieval-extinction effect. The importance of this point comes through at several places in the paper. E.g., the two groups in Study 1 experienced a different interval between the first and second CS extinction trials; and the results varied with this interval: a longer interval (10 min) ultimately resulted in less reinstatement of fear than a shorter interval. Even if the different pattern of results in these two groups was shown/known to imply two different processes, there is nothing in the present study that addresses what those processes might be. That is, while the authors talk about mechanisms of memory updating, there is little in the present study that permits any clear statement about mechanisms of memory. The references to a "short-term memory update" process do not help the reader to understand what is happening in the protocol.

We agree with the reviewer that whether and how the retrieval-extinction paradigm works is still under debate. Our results provide another line of evidence that such a paradigm is effective in producing long term fear amnesia. The focus of the current manuscript is to demonstrate that the retrieval-extinction paradigm can also facilitate a short-term fear memory deficit measured by SCR. Our TMS study provided some preliminary evidence in terms of the brain mechanisms involved in the causal relationship between the dorsolateral prefrontal cortex (dlPFC) activity and the short-term fear amnesia and showed that both the retrieval interval and the intact dlPFC activity were necessary for the short-term fear memory deficit and accordingly were referred to as the “mechanism” for memory update. We acknowledge that the term “mechanism” might have different connotations for different researchers. We now more explicitly clarify what we mean by “mechanisms” in the manuscript (line 99) as follows:

“In theory, different cognitive mechanisms underlying specific fear memory deficits, therefore, can be inferred based on the difference between memory deficits.”

In reply to this point, the authors cite evidence to suggest that "an isolated presentation of the CS+ seems to be important in preventing the return of fear expression." They then note the following: "It has also been suggested that only when the old memory and new experience (through extinction) can be inferred to have been generated from the same underlying latent cause, the old memory can be successfully modified (Gershman et al., 2017). On the other hand, if the new experiences are believed to be generated by a different latent cause, then the old memory is less likely to be subject to modification. Therefore, the way the 1stand 2ndCS are temporally organized (retrieval-extinction or standard extinction) might affect how the latent cause is inferred and lead to different levels of fear expression from a theoretical perspective." This merely begs the question: why might an isolated presentation of the CS+ result in the subsequent extinction experiences being allocated to the same memory state as the initial conditioning experiences? This is not yet addressed in any way.

As in our previous response, this manuscript is not about investigating the cognitive mechanism why and how an isolated presentation of the CS+ would suppress fear expression in the long term. As the reviewer is aware, and as we have addressed in our previous response letters, both the positive and negative evidence abounds as to whether the retrieval-extinction paradigm can successfully suppress the long-term fear expression. Previous research depicted mechanisms instigated by the single CS+ retrieval at the molecular, cellular, and systems levels, as well as through cognitive processes in humans. In the current manuscript, we simply set out to test that in addition to the long-term fear amnesia, whether the retrieval-extinction paradigm can also affect subjects’ short-term fear memory.

(2) The discussion of memory suppression is potentially interesting but, in its present form, raises more questions than it answers. That is, memory suppression is invoked to explain a particular pattern of results but I, as the reader, have no sense of why a fear memory would be better suppressed shortly after the retrieval-extinction protocol compared to the standard extinction protocol; and why this suppression is NOT specific to the cue that had been subjected to the retrieval-extinction protocol.

Memory suppression is the hypothesis we proposed that might be able to explain the results we obtained in the experiments. We discussed the possibility of memory suppression and listed the reasons why such a mechanism might be at work. As we mentioned in the manuscript, our findings are consistent with the memory suppression mechanism on at least two aspects: (1) cue-independence and (2) thought-control ability dependence. We agree that the questions raised by the reviewer are interesting but to answer these questions would require a series of further experiments to disentangle all the various variables and conceptual questions about the purpose of a phenomenon, which we are afraid is out of the scope of the current manuscript. We refer the reviewer to the discussion section where memory suppression might be the potential mechanism for the short-term amnesia we observed (lines 562-569) as follows:

“Previous studies indicate that a suppression mechanism can be characterized by three distinct features: first, the memory suppression effect tends to emerge early, usually 10-30 mins after memory suppression practice and can be transient (MacLeod and Macrae, 2001; Saunders and MacLeod, 2002); second, the memory suppression practice seems to directly act upon the unwanted memory itself (Levy and Anderson, 2002), such that the presentation of other cues originally associated with the unwanted memory also fails in memory recall (cue-independence); third, the magnitude of memory suppression effects is associated with individual difference in control abilities over intrusive thoughts (Küpper et al., 2014).”

(3) Relatedly, how does the retrieval-induced forgetting (which is referred to at various points throughout the paper) relate to the retrieval-extinction effect? The appeal to retrieval-induced forgetting as an apparent justification for aspects of the present study reinforces points 2 and 3 above. It is not uninteresting but lacks clarification/elaboration and, therefore, its relevance appears superficial at best.

We brought the topic of retrieval-induced forgetting (RIF) to stress the point that memory suppression can be unconscious. In a standard RIF paradigm, unlike the think/no-think paradigm, subjects are not explicitly told to suppress the non-target memories. However, to successfully retrieve the target memory, the cognitive system actively inhibits the non-target memories, effectively implementing a memory suppression mechanism (though unconsciously). Therefore, it is possible our results might be explained by the memory suppression framework. We elaborated this point in the discussion section (lines 578-584):

“In our experiments, subjects were not explicitly instructed to suppress their fear expression, yet the retrieval-extinction training significantly decreased short-term fear expression. These results are consistent with the short-term amnesia induced with the more explicit suppression intervention (Anderson et al., 1994; Kindt and Soeter, 2018; Speer et al., 2021; Wang et al., 2021; Wells and Davies, 1994). It is worth noting that although consciously repelling unwanted memory is a standard approach in memory suppression paradigm, it is possible that the engagement of the suppression mechanism can be unconscious.”

(4) I am glad that the authors have acknowledged the papers by Chalkia, van Oudenhove & Beckers (2020) and Chalkia et al (2020), which failed to replicate the effects of retrieval-extinction reported by Schiller et al in Reference 6. The authors have inserted the following text in the revised manuscript: "It should be noted that while our long-term amnesia results were consistent with the fear memory reconsolidation literature, there were also studies that failed to observe fear prevention (Chalkia, Schroyens, et al., 2020; Chalkia, Van Oudenhove, et al., 2020; Schroyens et al., 2023). Although the memory reconsolidation framework provides a viable explanation for the long-term amnesia, more evidence is required to validate the presence of reconsolidation, especially at the neurobiological level (Elsey et al., 2018). While it is beyond the scope of the current study to discuss the discrepancies between these studies, one possibility to reconcile these results concerns the procedure for the retrieval-extinction training. It has been shown that the eligibility for old memory to be updated is contingent on whether the old memory and new observations can be inferred to have been generated by the same latent cause (Gershman et al., 2017; Gershman and Niv, 2012). For example, prevention of the return of fear memory can be achieved through gradual extinction paradigm, which is thought to reduce the size of prediction errors to inhibit the formation of new latent causes (Gershman, Jones, et al., 2013). Therefore, the effectiveness of the retrieval-extinction paradigm might depend on the reliability of such paradigm in inferring the same underlying latent cause." Firstly, if it is beyond the scope of the present study to discuss the discrepancies between the present and past results, it is surely beyond the scope of the study to make any sort of reference to clinical implications!!!

As we have clearly stated in our manuscript that this paper was not about discussing why some literature was or was not able to replicate the retrieval-extinction results originally reported by Schiller et al. 2010. Instead, we aimed to report a novel short-term fear amnesia through the retrieval-extinction paradigm, above and beyond the long-term amnesia reported before. Speculating about clinical implications of these finding is unrelated to the long-term, amnesia debate in the reconsolidation world. We now refer the reader to several perspectives and reviews that have proposed ways to resolve these discrepancies as follows (lines 642-673).

Secondly, it is perfectly fine to state that "the effectiveness of the retrieval-extinction paradigm might depend on the reliability of such paradigm in inferring the same underlying latent cause..." This is not uninteresting, but it also isn't saying much. Minimally, I would expect some statement about factors that are likely to determine whether one is or isn't likely to see a retrieval-extinction effect, grounded in terms of this theory.

Again, as we have responded many times, we simply do not know why some studies were able to suppress the fear expression using the retrieval-extinction paradigm and other studies weren’t. This is still an unresolved issue that the field is actively engaging with, and we now refer the reader to several papers dealing with this issue. However, this is NOT the focus of our manuscript. Having a healthy debate does not mean that every study using the retrieval-extinction paradigm must address the long-standing question of why the retrieval-extinction paradigm is effective (at least in some studies).

Clarifications, Elaborations, Edits(5) Some parts of the paper are not easy to follow. Here are a few examples (though there are others):(a) In the abstract, the authors ask "whether memory retrieval facilitates update mechanisms other than memory reconsolidation"... but it is never made clear how memory retrieval could or should "facilitate" a memory update mechanism.

We meant to state that the retrieval-extinction paradigm might have effects on fear memory, above and beyond the purported memory reconsolidation effect. Sentence modified (lines 25-26) as follows:

“Memory reactivation renders consolidated memory fragile and thereby opens the window for memory updates, such as memory reconsolidation.”

(b) The authors state the following: "Furthermore, memory reactivation also triggers fear memory reconsolidation and produces cue specific amnesia at a longer and separable timescale (Study 2, N = 79 adults)." Importantly, in study 2, the retrieval-extinction protocol produced a cue-specific disruption in responding when testing occurred 24 hours after the end of extinction. This result is interesting but cannot be easily inferred from the statement that begins "Furthermore..." That is, the results should be described in terms of the combined effects of retrieval and extinction, not in terms of memory reactivation alone; and the statement about memory reconsolidation is unnecessary. One can simply state that the retrieval-extinction protocol produced a cue-specific disruption in responding when testing occurred 24 hours after the end of extinction.

The sentence the reviewer referred to was in our original manuscript submission but had since been modified based on the reviewer’s comments from last round of revision. Please see the abstract (lines 30-35) of our revised manuscript from last round of revision:

“Furthermore, across different timescales, the memory retrieval-extinction paradigm triggers distinct types of fear amnesia in terms of cue-specificity and cognitive control dependence, suggesting that the short-term fear amnesia might be caused by different mechanisms from the cue-specific amnesia at a longer and separable timescale (Study 2, *N* = 79 adults).”

(c) The authors also state that: "The temporal scale and cue-specificity results of the short-term fear amnesia are clearly dissociable from the amnesia related to memory reconsolidation, and suggest that memory retrieval and extinction training trigger distinct underlying memory update mechanisms." ***The pattern of results when testing occurred just minutes after the retrieval-extinction protocol was different to that obtained when testing occurred 24 hours after the protocol. Describing this in terms of temporal scale is unnecessary; and suggesting that memory retrieval and extinction trigger different memory update mechanisms is not obviously warranted. The results of interest are due to the combined effects of retrieval+extinction and there is no sense in which different memory update mechanisms should be identified with the different pattern of results obtained when testing occurred either 30 min or 24 hours after the retrieval-extinction protocol (at least, not the specific pattern of results obtained here).

Again, we are afraid that the reviewer referred to the abstract in the original manuscript submission, instead of the revised abstract we submitted in the last round. Please see lines 37-39 of the revised abstract where the sentence was already modified (or the abstract from last round of revision).

The facts that the 30min, 6hr and 24hr test results are different in terms of their cue-specificity and thought-control ability dependence are, to us, an important discovery in terms of delineating different cognitive processes at work following the retrieval-extinction paradigm. We want to emphasize that the fear memories after going through the retrieval-extinction paradigm showed interesting temporal dynamics in terms of their magnitudes, cue-specificity and thought-control ability dependence.

(d) The authors state that: "We hypothesize that the labile state triggered by the memory retrieval may facilitate different memory update mechanisms following extinction training, and these mechanisms can be further disentangled through the lens of temporal dynamics and cue-specificities." *** The first part of the sentence is confusing around usage of the term "facilitate"; and the second part of the sentence that references a "lens of temporal dynamics and cue-specificities" is mysterious. Indeed, as all rats received the same retrieval-extinction exposures in Study 2, it is not clear how or why any differences between the groups are attributed to "different memory update mechanisms following extinction"

The term “facilitate” was used to highlight the fact that the short-term fear amnesia effect is also memory retrieval dependent, as study 1 demonstrated. The novelty of the short-term fear memory deficit can be distinguished from the long-term memory effect via cue-specificity and thought-control ability dependence. Sentence has been modified (lines 97-101) as follows:

“We hypothesize that the labile state triggered by the memory retrieval may facilitate different memory deficits following extinction training, and these deficits can be further disentangled through the lens of temporal dynamics and cue-specificities. In theory, different cognitive mechanisms underlying specific fear memory deficits, therefore, can be inferred based on the difference between memory deficits.”

Data(6A) The eight participants who were discontinued after Day 1 in Study 1 were all from the no reminder group. The authors should clarify how participants were allocated to the two groups in this experiment so that the reader can better understand why the distribution of non-responders was non-random (as it appears to be).(6B) Similarly, in study 2, of the 37 participants that were discontinued after Day 2, 19 were from Group 30 min and 5 were from Group 6 hours. The authors should comment on how likely these numbers are to have been by chance alone. I presume that they reflect something about the way that participants were allocated to groups: e.g., the different groups of participants in studies 1 and 2 could have been run at quite different times (as opposed to concurrently). If this was done, why was it done? I can't see why the study should have been conducted in this fashion - this is for myriad reasons, including the authors' concerns re SCRs and their seasonal variations.

As we responded in the previous response letters (as well as in the revised the manuscript), subjects were excluded because their SCR did not reach the threshold of 0.02 S when electric shock was applied. Subjects were assigned to different treatments daily (eg. Day 1 for the reminder group and Day 2 for no-reminder group) to avoid potential confusion in switching protocols to different subjects within the same day. We suspect that the non-responders might be related to the body thermal conditions caused by the lack of central heating for specific dates. Please note that the discontinued subjects (non-responders) were let go immediately after the failure to detect their SCR (< 0.02 S) on Day 1 and never invited back on Day 2, so it’s possible that the discontinued subjects were all from certain dates on which the body thermal conditions were not ideal for SCR collection. Despite the number of excluded subjects, we verified the short-term fear amnesia effect in three separate studies, which to us should serve as strong evidence in terms of the validity of the effect.

(6C) In study 2, why is responding to the CS- so high on the first test trial in Group 30 min? Is the change in responding to the CS- from the last extinction trial to the first test trial different across the three groups in this study? Inspection of the figure suggests that it is higher in Group 30 min relative to Groups 6 hours and 24 hours. If this is confirmed by the analysis, it has implications for the fear recovery index which is partly based on responses to the CS-. If not for differences in the CS- responses, Groups 30 min and 6 hours are otherwise identical. That is, the claim of differential recovery to the CS1 and CS2 across time may simply an artefact of the way that the recovery index was calculated. This is unfortunate but also an important feature of the data given the way in which the fear recovery index was calculated.

We have provided detailed analysis to this question in our previous response letter, and we are posting our previous response there:

Following the reviewer’s comments, we went back and calculated the mean SCR difference of CS- between the first test trial and the last extinction trial for all three studies (see Author response image 1 below). In study 1, there was no difference in the mean CS- SCR (between the first test trial and last extinction trial) between the reminder and no-reminder groups Kruskal-Wallis test \begin{document}$\chi_{1}^{2}=0.230, P=0.631$\end{document}, though both groups showed significant fear recovery even in the CS- condition (Wilcoxon signed rank test, reminder: *P* = 0.0043, no-reminder: *P* = 0.0037). Next, we examined the mean SCR for CS- for the 30min, 6h and 24h groups in study 2 and found that there was indeed a group difference (one-way ANOVA,*F*_2.76_ = 5.3462, *P* = 0.0067, panel b), suggesting that the CS- related SCR was influenced by the test time (30min, 6h or 24h). We also tested the CS- related SCR for the 4 groups in study 3 (where test was conducted 1 hour after the retrieval-extinction training) and found that across TMS stimulation types (PFC vs. VER) and reminder types (reminder vs. no-reminder) the ANOVA analysis did not yield main effect of TMS stimulation type (*F*_1.71_ = 0.322, *P* = 0.572) nor main effect of reminder type (*F*_1.71_ = 0.0499, *P* = 0.824, panel c). We added the R-VER group results in study 3 (see panel c) to panel b and plotted the CS- SCR difference across 4 different test time points and found that CS- SCR decreased as the test-extinction delay increased (Jonckheere-Terpstra test, *P* = 0.00028). These results suggest a natural “forgetting” tendency for CS- related SCR and highlight the importance of having the CS- as a control condition to which the CS+ related SCR was compared with.

(6D) The 6 hour group was clearly tested at a different time of day compared to the 30 min and 24 hour groups. This could have influenced the SCRs in this group and, thereby, contributed to the pattern of results obtained.

Again, we answered this question in our previous response. Please see the following for our previous response:

For the 30min and 24h groups, the test phase can be arranged in the morning, in the afternoon or at night. However, for the 6h group, the test phase was inevitably in the afternoon or at night since we wanted to exclude the potential influence of night sleep on the expression of fear memory (see Author response table 1 below). If we restricted the test time in the afternoon or at night for all three groups, then the timing of their extinction training was not matched.

**Author response table 1. sa3table1:** 

Testtime/Group	Morning	Afternoon	Night	Total# Subjects
30 min	10	9	8	27
6 h	0	12	14	26
24 h	8	9	9	26

Nevertheless, we also went back and examined the data for the subjects only tested in the afternoon or at nights in the 30min and 24h groups to match with the 6h group where all the subjects were tested either in the afternoon or at night. According to the table above, we have 17 subjects for the 30min group (9+8),18 subjects for the 24h group (9 + 9) and 26 subjects for the 6h group (12 + 14). As Author response image 2 shows, the SCR patterns in the fear acquisition, extinction and test phases were similar to the results presented in the original figure.

**Author response image 2. sa3fig2:** 

(6E) The authors find different patterns of responses to CS1 and CS2 when they were tested 30 min after extinction versus 24 h after extinction. On this basis, they infer distinct memory update mechanisms. However, I still can't quite see why the different patterns of responses at these two time points after extinction need to be taken to infer different memory update mechanisms. That is, the different patterns of responses at the two time points could be indicative of the same "memory update mechanism" in the sense that the retrieval-extinction procedure induces a short-term memory suppression that serves as the basis for the longer-term memory suppression (i.e., the reconsolidation effect). My pushback on this point is based on the notion of what constitutes a memory update mechanism; and is motivated by what I take to be a rather loose use of language/terminology in the reconsolidation literature and this paper specifically (for examples, see the title of the paper and line 2 of the abstract).

As we mentioned previously, the term “mechanism” might have different connotations for different researchers. We aim to report a novel memory deficit following the retrieval-extinction paradigm, which differed significantly from the purported reconsolidation related long-term fear amnesia in terms of its timescale, cue-specificity and thought-control ability. Further TMS study confirmed that the intact dlPFC function is necessary for the short-term memory deficit. It’s based on these results we proposed that the short-term fear amnesia might be related to a different cognitive “mechanism”. As mentioned above, we now clarify what we mean by “mechanism” in the abstract and introduction (lines 31-34, 97-101).

**Reviewer #2 (Public review):**
The fear acquisition data is converted to a differential fear SCR and this is what is analysed (early vs late). However, the figure shows the raw SCR values for CS+ and CS- and therefore it is unclear whether acquisition was successful (despite there being an "early" vs "late" effect - no descriptives are provided).(1) There are still no descriptive statistics to substantiate learning in Experiment 1.

We answered this question in our previous response letter. We are sorry that the definition of “early” and “late” trials was scattered in the manuscript. For example, we wrote “the late phase of acquisition (last 5 trials)” (Line 375-376) in the results section. Since there were 10 trials in total for the acquisition stage, we define the first 5 trials and the last 5 trials as “early” and “late” phases of the acquisition stage and explicitly added them into the first occasion “early” and “late” terms appeared (lines 316-318).

In the results section, we did test whether the acquisition was successful in our previous manuscript (Line 316-325):

“To assess fear acquisition across groups (Figure 1B and C), we conducted a mixed two-way ANOVA of group (reminder vs. no-reminder) x time (early vs. late part of the acquisition; first 5 and last 5 trials, correspondingly) on the differential fear SCR. Our results showed a significant main effect of time (early vs. late; *F*_1,55_ = 6.545, *P* = 0.013, *η*^2^ = 0.106), suggesting successful fear acquisition in both groups. There was no main effect of group (reminder vs. no-reminder) or the group x time interaction (group: *F*_1,55_ = 0.057, *P* = 0.813, *η*^2^ = 0.001; interaction: *F*_1,55_ = 0.066, *P* = 0.798, *η*^2^ = 0.001), indicating similar levels of fear acquisition between two groups. Post-hoc *t*-tests confirmed that the fear responses to the CS+ were significantly higher than that of CS- during the late part of acquisition phase in both groups (reminder group: *t*_29_ = 6.642, *P* < 0.001; no-reminder group: *t*_26_ = 8.522, *P* < 0.001; Figure 1C). Importantly, the levels of acquisition were equivalent in both groups (early acquisition: *t*_55_ = -0.063, *P* = 0.950; late acquisition: *t*_55_ = -0.318, *P* = 0.751; Figure 1C).”

In Experiment 1 (Test results) it is unclear whether the main conclusion stems from a comparison of the test data relative to the last extinction trial ("we defined the fear recovery index as the SCR difference between the first test trial and the last extinction trial for a specific CS") or the difference relative to the CS- ("differential fear recovery index between CS+ and CS-"). It would help the reader assess the data if Fig 1e presents all the indexes (both CS+ and CS-). In addition, there is one sentence which I could not understand "there is no statistical difference between the differential fear recovery indexes between CS+ in the reminder and no reminder groups (P=0.048)". The p value suggests that there is a difference, yet it is not clear what is being compared here. Critically, any index taken as a difference relative to the CS- can indicate recovery of fear to the CS+ or absence of discrimination relative to the CS-, so ideally the authors would want to directly compare responses to the CS+ in the reminder and no-reminder groups. In the absence of such comparison, little can be concluded, in particular if SCR CS- data is different between groups. The latter issue is particularly relevant in Experiment 2, in which the CS- seems to vary between groups during the test and this can obscure the interpretation of the result.(2) In the revised analyses, the authors now show that CS- changes in different groups (for example, Experiment 2) so this means that there is little to conclude from the differential scores because these depend on CS-. It is unclear whether the effects arise from CS+ performance or the differential which is subject to CS- variations.

There was a typo in the “P = 0.048” sentence and we have corrected it in our last response letter. Also in the previous response letter, we specifically addressed how the fear recovery index was defined (also in the revised manuscript).

In most of the fear conditioning studies, CS- trials were included as the baseline control. In turn, most of the analyses conducted also involved comparisons between different groups. Directly comparing CS+ trials across groups (or conditions) is rare. In our study 2, we showed that the CS- response decreased as a function of testing delays (30min, 1hr, 6hr and 24hr). Ideally, it would be nice to show that the CS- across groups/conditions did not change. However, even in those circumstances, comparisons are still based on the differential CS response (CS+ minus CS-), that is, the difference of difference. It is also important to note that difference score is important as CS+ alone or across conditions is difficult to interpret, especially in humans, due to noise, signal fluctuations, and irrelevant stimulus features; therefore trials-wise reference is essential to assess the CS+ in the context of a reference stimulus in each trial (after all, the baselines are different). We are listing a few influential papers in the field that the CS- responses were not particularly equivalent across groups/conditions and argue that this is a routine procedure (Kindt & Soeter 2018 Figs. 2-3; Sevenster et al., 2013 Fig. 3; Liu et al., 2014 Fig. 1; Raio et al., 2017 Fig. 2).

In experiment 1, the findings suggest that there is a benefit of retrieval followed by extinction in a short-term reinstatement test. In Experiment 2, the same effect is observed to a cue which did not undergo retrieval before extinction (CS2+), a result that is interpreted as resulting from cue-independence, rather than a failure to replicate in a within-subjects design the observations of Experiment 1 (between-subjects). Although retrieval-induced forgetting is cue-independent (the effect on items that are suppressed [Rp-] can be observed with an independent probe), it is not clear that the current findings are similar, and thus that the strong parallels made are not warranted. Here, both cues have been extinguished and therefore been equally exposed during the critical stage.(3) The notion that suppression is automatic is speculative at best

We have responded the same question in our previous revision. Please note that our results from study 1 (the comparison between reminder and no-reminder groups) was not set up to test the cue-independence hypothesis for the short-term amnesia with only one CS+. Results from both study 2 (30min condition) and study 3 confirmed the cue-independence hypothesis and therefore we believe interpreting results from study 2 as “a failure to replicate in a within-subject design of the observations of Experiment 1” is not the case.

We agree that the proposal of automatic or unconscious memory suppression is speculative and that’s why we mentioned it in the discussion. The timescale, cue-specificity and the thought-control ability dependence of the short-term fear amnesia identified in our studies was reminiscent of the memory suppression effects reported in the previous literature. However, memory suppression typically adopted a conscious “suppression” treatment (such as the think/no-think paradigm), which was absent in the current study. However, the retrieval-induced forgetting (RIF), which is also considered a memory suppression paradigm via inhibitory control, does not require conscious effort to suppress any particular thought. Based on these results and extant literature, we raised the possibility of memory suppression as a potential mechanism. We make clear in the discussion that the suppression hypothesis and connections with RIF will require further evidence (lines 615-616):

“future research will be needed to investigate whether the short-term effect we observed is specifically related to associative memory or the spontaneous nature of suppression as in RIF (Figure 6C).”

(4) It still struggle with the parallels between these findings and the "limbo" literature. Here you manipulated the retention interval, whereas in the cited studies the number of extinction (exposure) was varied. These are two completely different phenomena.

We borrowed the “limbo” term to stress the transitioning from short-term to long-term memory deficits (the 6hr test group). Merlo et al. (2014) found that memory reconsolidation and extinction were dissociable processes depending on the extent of memory retrieval. They argued that there was a “limbo” transitional state, where neither the reconsolidation nor the extinction process was engaged. Our results suggest that at the test delay of 6hr, neither the short-term nor the long-term effect was present, signaling a “transitional” state after which the short-term memory deficit wanes and the long-term deficit starts to take over. We make this idea more explicit as follows (lines 622-626):

“These works identified important “boundary conditions” of memory retrieval in affecting the retention of the maladaptive emotional memories. In our study, however, we showed that even within a boundary condition previously thought to elicit memory reconsolidation, mnemonic processes other than reconsolidation could also be at work, and these processes jointly shape the persistence of fear memory.”

(5) My point about the data problematic for the reconsolidation (and consolidation) frameworks is that they observed memory in the absence of the brain substrates that are needed for memory to be observed. The answer did not address this. I do not understand how the latent cause model can explain this, if the only difference is the first ITI. Wouldn't participants fail to integrate extinction with acquisition with a longer ITI?

We take the sentence “they observed memory in the absence of the brain substrates that are needed for memory to be observed” as referring to the long-term memory deficit in our study. As we responded before, the aim of this manuscript was not about investigating the brain substrates involved in memory reconsolidation (or consolidation). Using a memory retrieval-extinction paradigm, we discovered a novel short-term memory effect, which differed from the purported reconsolidation effect in terms of timescale, cue-specificity and thought-control ability dependence. We further showed that both memory retrieval and intact dlPFC functions were necessary to observe the short-term memory deficit effect. Therefore, we conclude that the brain mechanism involved in such an effect should be different from the one related to the purported reconsolidation effect. We make this idea more explicit as follows (lines 546-547):

“Therefore, findings of the short-term fear amnesia suggest that the reconsolidation framework falls short to accommodate this more immediate effect (Figure 6A and B).”

Whilst I could access the data in the OFS site, I could not make sense of the Matlab files as there is no signposting indicating what data is being shown in the files. Thus, as it stands, there is no way of independently replicating the analyses reported.(6) The materials in the OSF site are the same as before, they haven't been updated.

Last time we thought the main issue was the OSF site not being publicly accessible and thus made it open to all visitors. We have added descriptive file to explain the variables to help visitors to replicate the analyses we took.

(7) Concerning supplementary materials, the robustness tests are intended to prove that you (1) can get the same results by varying the statistical models or (2) you can get the same results when you include all participants. Here authors have done both so this does not help. Also, in the rebuttal letter, they stated "Please note we did not include non-learners in these analyses " which contradicts what is stated in the figure captions "(learners + non learners)"

In the supplementary materials, we did the analyses of varying the statistical models and including both learners and non-learners separately, instead of both. In fact, in the supplementary material Figs. 1 & 2, we included all the participants and performed similar analysis as in the main text and found similar results (learners + non-learners). Also, in the text of the supplementary material, we used a different statistical analysis method to only learners (analyzing subjects reported in the main text using a different method) and achieved similar results. We believe this is exactly what the reviewer suggested us to do. Also there seems to be a misunderstanding for the "Please note we did not include non-learners in these analyses" sentence in the rebuttal letter. As the reviewer can see, the full sentence read “Please note we did not include non-learners in these analyses (the texts of the supplementary materials)”. We meant to express that the Figures and texts in the supplementary material reflect two approaches: (1) Figures depicting re-analysis with all the included subjects (learners + non learners); (2) Text describing different analysis with learners. We added clarifications to emphasize these approaches in the supplementary materials.

(8) Finally, the literature suggesting that reconsolidation interference "eliminates" a memory is not substantiated by data nor in line with current theorising, so I invite a revision of these strong claims.

We agree and have toned down the strong claims.

Overall, I conclude that the revised manuscript did not address my main concerns.

In both rounds of responses, we tried our best to address the reviewer’s concerns. We hope that the clarifications in this letter and revisions in the text address the remaining concerns. Thank you for your feedback.

Reference:

Kindt, M. and Soeter, M. 2018. Pharmacologically induced amnesia for learned fear is time and sleep dependent. Nat Commun, 9, 1316.

Liu, J., Zhao, L., Xue, Y., Shi, J., Suo, L., Luo, Y., Chai, B., Yang, C., Fang, Q., Zhang, Y., Bao, Y., Pickens, C. L. and Lu, L. 2014. An unconditioned stimulus retrieval extinction procedure to prevent the return of fear memory. Biol Psychiatry, 76, 895-901.

Raio, C. M., Hartley, C. A., Orederu, T. A., Li, J. and Phelps, E. A. 2017. Stress attenuates the flexible updating of aversive value. Proc Natl Acad Sci U S A, 114, 11241-11246.

Sevenster, D., Beckers, T., & Kindt, M. 2013. Prediction error governs pharmacologically induced amnesia for learned fear. Science (New York, N.Y.), 339(6121), 830–833.